# Heavy-Tailed Linear Bandits: Huber Regression with One-Pass Update

**Jing Wang** [1 2]  **Yu-Jie Zhang** [3]  **Peng Zhao** [1 2]  **Zhi-Hua Zhou** [1 2]

## Abstract

We study the stochastic linear bandits with heavy-tailed noise. Two principled strategies for handling heavy-tailed noise, truncation and median-of-means, have been introduced to heavy-tailed bandits. Nonetheless, these methods rely on specific noise assumptions or bandit structures, limiting their applicability to general settings. The recent work (Huang et al., 2023) develops a soft truncation method via the adaptive Huber regression to address these limitations. However, their method suffers undesired computational costs: it requires storing all historical data and performing a full pass over these data at each round. In this paper, we propose a *one-pass* algorithm based on the online mirror descent framework. Our method updates using only current data at each round, reducing the per-round computational cost from $\mathcal{O}(t \log T)$ to $\mathcal{O}(1)$ with respect to current round $t$ and the time horizon $T$, and achieves a near-optimal and variance-aware regret of order $\widetilde{\mathcal{O}}\big(dT^{\frac{1-\varepsilon}{2(1+\varepsilon)}}\sqrt{\sum_{t=1}^{T}\nu_t^2} + dT^{\frac{1-\varepsilon}{2(1+\varepsilon)}}\big)$ where $d$ is the dimension and $\nu_t^{1+\varepsilon}$ is the $(1+\varepsilon)$-th central moment of reward at round $t$.

## 1. Introduction

Stochastic Linear Bandits (SLB) models sequential decision-making process with linear structured reward distributions, which has been extensively studied in the literature (Dani et al., 2008; Abbasi-Yadkori et al., 2011; Li et al., 2021b). Specifically, in SLB the observed reward at time $t$ is the inner product of the arm's feature vector $X_t$ and an unknown parameter $\theta_*$, corrupted by sub-Gaussian noise $\eta_t$, namely,

$$r_t = X_t^\top \theta_* + \eta_t. \tag{1}$$

However, in many real-world scenarios, the noise in data may follow a heavy-tailed distribution, such as in financial markets (Hull, 2012) and online advertising (Jebarajakirthy et al., 2021). This motivates the study of heavy-tailed linear bandits (HvtLB) (Medina & Yang, 2016; Shao et al., 2018; Xue et al., 2020; Huang et al., 2023), where the noise $\eta_t$ in (1) satisfies that for some $\varepsilon \in (0, 1]$, $\nu > 0$,

$$\mathbb{E}[\eta_t] = 0, \quad \mathbb{E}[|\eta_t|^{1+\varepsilon}] \le \nu^{1+\varepsilon}. \tag{2}$$

To address heavy-tailed noise, statistical methods in estimation and regression often rely on two key principles: truncation and median-of-means (MOM) (Lugosi & Mendelson, 2019). Truncation methods handle outliers by directly removing extreme data points. Soft version truncation, such as Catoni's M-estimator (Catoni, 2012), reduces the impact of outliers by assigning them lower weights. This ensures robust estimates while maintaining potentially valuable information from extreme data points. Median-of-means (MOM) methods take a different approach by dividing the dataset into several groups, calculating the mean within each group, and then using the median of these group means as the final estimate. This method limits the influence of outliers to only a few groups, preventing impact on the overall dataset.

For HvtLB, Shao et al. (2018) developed algorithms using truncation and MOM-based least squares approach for parameter estimation, and achieved an $\widetilde{\mathcal{O}}\big(dT^{\frac{1}{1+\varepsilon}}\big)$ regret bound which is proven to be optimal. However, these approaches have some notable issues: truncation methods require the assumption of bounded absolute moments for rewards, which cannot vanish in deterministic case, making them suboptimal in noiseless scenarios; MOM methods rely on repeated arm pulling and assumption of a fixed arm set, which makes them difficult to extend to more general settings. These limitations reveal that methods based on truncation and MOM heavily depend on specific assumptions or the bandit structure. For broad decision-making scenarios, such as online MDPs and adaptive control, where the state or context may vary dynamically, these assumptions no longer hold. A more detailed discussion of these challenges is provided in Section 6.

Recently, Sun et al. (2020) made remarkable progress in the study of heavy-tailed statistics by proposing adaptive Huber regression, which leverages the Huber loss (Huber, 1964) to achieve non-asymptotic guarantees. Li &

---

[1]National Key Laboratory for Novel Software Technology, Nanjing University, China [2]School of Artificial Intelligence, Nanjing University, China [3]Center for Advanced Intelligence Project, RIKEN, Japan. Correspondence to: Peng Zhao <zhaop@lamda.nju.edu.cn>.

*Proceedings of the 42nd International Conference on Machine Learning*, Vancouver, Canada. PMLR 267, 2025. Copyright 2025 by the author(s).

Table 1: Comparisons of our regret bounds and computational complexity to previous best-known results for heavy-tailed linear bandits. For the regret, the logarithmic dependence over $T$ is hidden by $\widetilde{\mathcal{O}}(\cdot)$-notation; $d$ is the dimension and $\nu_t$ is the moment bound. For the computational cost, we only keep the dependence on the time step $t$ and overall time step $T$.

| Method | Algorithm | Regret | Comp. cost | Remark |
|---|---|---|---|---|
| MOM | MENU (Shao et al., 2018) 
 CRMM (Xue et al., 2023) | $\widetilde{\mathcal{O}}\left(dT^{\frac{1}{1+\varepsilon}}\right)$ | $\mathcal{O}(\log T)$ 
 $\mathcal{O}(1)$ | fixed arm set and 
 repeated pulling |
| Truncation | TOFU (Shao et al., 2018) 
 CRTM (Xue et al., 2023) | $\widetilde{\mathcal{O}}\left(dT^{\frac{1}{1+\varepsilon}}\right)$ | $\mathcal{O}(t)$ 
 $\mathcal{O}(1)$ | absolute moment 
 $\mathbb{E}[\lvert r_t\rvert^{1+\varepsilon} \mid \mathcal{F}_{t-1}] \leq u$ |
| Huber | HEAVY-OFUL (Huang et al., 2023) | $\widetilde{\mathcal{O}}\left(dT^{\frac{1-\varepsilon}{2(1+\varepsilon)}}\sqrt{\sum_{t=1}^{T}\nu_t^2} + dT^{\frac{1-\varepsilon}{2(1+\varepsilon)}}\right)$ | $\mathcal{O}(t\log T)$ | instance-dependent bound |
| Huber | Hvt-UCB (Corollary 1) | $\widetilde{\mathcal{O}}\left(dT^{\frac{1}{1+\varepsilon}}\right)$ | $\mathcal{O}(1)$ | $\mathbb{E}[\lvert \eta_t\rvert^{1+\varepsilon} \mid \mathcal{F}_{t-1}] \leq \nu^{1+\varepsilon}$ |
| Huber | Hvt-UCB (Theorem 1) | $\widetilde{\mathcal{O}}\left(dT^{\frac{1-\varepsilon}{2(1+\varepsilon)}}\sqrt{\sum_{t=1}^{T}\nu_t^2} + dT^{\frac{1-\varepsilon}{2(1+\varepsilon)}}\right)$ | $\mathcal{O}(1)$ | instance-dependent bound |

Sun (2024) further applied this technique to SLB, achieving an optimal and variance-aware regret bound under bounded variance conditions ($\varepsilon = 1$). Subsequently, Huang et al. (2023) extended the adaptive Huber regression to handle heavy-tailed scenarios ($\varepsilon \in (0,1]$), achieving an optimal and instance-dependent regret bound of $\widetilde{\mathcal{O}}\big(dT^{\frac{1-\varepsilon}{2(1+\varepsilon)}}\sqrt{\sum_{t=1}^{T}\nu_t^2} + dT^{\frac{1-\varepsilon}{2(1+\varepsilon)}}\big)$ for HvtLB. Notably, the adaptive Huber regression does not rely on additional noise assumptions or the repeated pulling of bandit settings. This generality allows the approach to be further adapted to other decision-making scenarios, such as reinforcement learning with function approximation.

In each round $t$, the adaptive Huber regression has to solve the following optimization problem:

$$\widehat{\theta}_t = \underset{\theta \in \Theta}{\arg\min}\ \lambda \lVert\theta\rVert_2^2 + \sum_{s=1}^{t-1} \ell_s(\theta), \qquad (3)$$

where $\Theta$ is the feasible set of the parameter, $\lambda$ is the regularization parameter, and $\ell_s(\theta)$ represents the Huber loss at round $s$. Unlike the OFUL algorithm (Abbasi-Yadkori et al., 2011) for SLB with sub-Gaussian noise, which uses least squares (LS) for estimation and can update recursively based on the closed-form solution, the Huber loss is partially quadratic and partially linear. Solving (3) requires storing all historical data and performing a full pass over all historical data at each round $t$ to update the parameter estimation. This results in a per-round storage cost of $\mathcal{O}(t)$ and computational cost of $\mathcal{O}(t\log T)$ with respect to the current round $t$ and the time horizon $T$, making it computationally infeasible for large-scale or real-time applications. Thus, we require a *one-pass* algorithm for HvtLB that updates the estimation at each round $t$ using only the current data, without storing historical data. Indeed, our goal is to design an algorithm for heavy-tailed linear bandits that is (basically) as efficient as OFUL in SLB with sub-Gaussian noise. Note

that while one-pass algorithms based on the truncation and MOM approach have been proposed by Xue et al. (2023), but as noted earlier, these methods heavily rely on specific assumptions or the bandit structure. Given the attractive properties of Huber loss-based method (Huang et al., 2023), a natural question arises: *Is it possible to design a one-pass Huber-loss-based algorithm that achieves optimal regret?*

**Our Results.** In this work, we propose a Huber loss-based one-pass algorithm Hvt-UCB utilizing the Online Mirror Descent (OMD) framework, which achieves the optimal regret bound of $\widetilde{\mathcal{O}}(dT^{\frac{1}{1+\varepsilon}})$ while eliminating the need to store historical data. Additionally, our algorithm can further achieve the instance-dependent regret bound of $\widetilde{\mathcal{O}}\big(dT^{\frac{1-\varepsilon}{2(1+\varepsilon)}}\sqrt{\sum_{t=1}^{T}\nu_t^2} + dT^{\frac{1-\varepsilon}{2(1+\varepsilon)}}\big)$, where $\nu_t^{1+\varepsilon}$ is the $(1+\varepsilon)$-th central moment of reward at round $t$. Our approach preserves both optimal and instance-dependent regret guarantees achieved by Huang et al. (2023) while significantly reducing the per-round computational cost from $\mathcal{O}(t\log T)$ to $\mathcal{O}(1)$ with respect to the current round $t$ and the time horizon $T$. Moreover, our algorithm does not rely on additional assumptions, making it more broadly applicable. A comprehensive comparison of our results and previous works on heavy-tailed linear bandits is presented in Table 1.

**Key Challenges and Techniques.** Designing a one-pass algorithm for heavy-tailed linear bandits presents two major challenges. On the one hand, unlike the full-batch estimator (3) (Huang et al., 2023), which leverages the entire historical data to control the estimation error, a one-pass estimator updates relies only on the current data point. Achieving comparable estimation error bound for one-pass estimator is much more challenging, especially in the presence of heavy-tailed noise. On the other hand, OMD is originally developed for regret minimization. Adapting it to parameter estimation requires linking the parameter gap (estimation

error) with the loss value gap (regret). To address these challenges, our algorithm design and analysis incorporate two key components: *(i) Handling heavy-tailed noise:* Zhang et al. (2016) have adopted a variant of Online Newton Step (ONS) to Logistic Bandits settings, but their method cannot be applied to HvtLB since their method relies on the global strong convexity of the logistic loss, whereas the Huber loss is partially linear. We generalize their approach to the OMD framework and introduce a recursive normalization factor that ensures normal data stays in the quadratic region of the Huber loss with high probability, maintaining robustness in estimation. This normalization factor also introduces a negative term in the estimation error, which is crucial for controlling stability. *(ii) Supporting one-pass updates:* The OMD framework introduces a stability term in estimation error, which represents the impact of online updates. The adaptive normalization factor adds a multiplicative bias to this term, resulting in a positive term of order $\widetilde{\mathcal{O}}(\sqrt{T})$ that weakens the bound. This requires a careful design of the normalization factor and learning rate of OMD to cancel it out using the negative term that arises from our OMD-based analysis, which is similar to a crucial technique in the regret analysis of recent gradient-variation online learning methods (Zhao et al., 2020b; 2024).

## 2. Related Work

**Heavy-Tailed Bandits.** The multi-armed bandits (MAB) problem with heavy-tailed rewards, characterized by finite $(1 + \varepsilon)$-moment, was first studied by Bubeck et al. (2013). They introduced two widely used robust estimation and regression methods from statistics into the heavy-tailed bandits setting: truncation and median-of-means (MOM). Medina & Yang (2016) extended these methods to heavy-tailed linear bandits (HvtLB) and proposed two algorithms: the truncation-based CRT algorithm and the MOM-based MoM algorithm, achieving $\widetilde{\mathcal{O}}\big(dT^{\frac{2+\varepsilon}{2(1+\varepsilon)}}\big)$ and $\widetilde{\mathcal{O}}\big(dT^{\frac{1+2\varepsilon}{1+3\varepsilon}}\big)$ regret bounds, respectively. Later, Shao et al. (2018) established an $\Omega\big(dT^{\frac{1}{1+\varepsilon}}\big)$ lower bound for HvtLB and introduced the truncation-based TOFU algorithm and the MOM-based MENU algorithm, both achieving an $\widetilde{\mathcal{O}}\big(dT^{\frac{1}{1+\varepsilon}}\big)$ regret bound. For HvtLB with finite arm set, Xue et al. (2020) established an $\Omega\big(d^{\frac{\varepsilon}{1+\varepsilon}}T^{\frac{1}{1+\varepsilon}}\big)$ lower bound, and proposed the truncation-based BTC algorithm and MOM-based BMM algorithm, both of which achieve an $\widetilde{\mathcal{O}}\big(\sqrt{d}T^{\frac{1}{1+\varepsilon}}\big)$ regret bound, reducing the dependence on the dimension $d$ under the finite arm set setting. To the best of our knowledge, the two works (Kang & Kim, 2023; Li & Sun, 2024) first introduce Huber loss to linear bandits. Specifically, Kang & Kim (2023) studied the heavy-tailed linear contextual bandits (HvtLCB) with fixed and finite arm set, where the context $X_t$ at each round is independently and identically distributed from a fixed distribution. They proposed the

Huber-Bandit algorithm based on Huber regression, achieving an $\widetilde{\mathcal{O}}\big(\sqrt{d}T^{\frac{1}{1+\varepsilon}}\big)$ regret bound, On the other hand, Li & Sun (2024) were the first to apply adaptive Huber regression technique (Sun et al., 2020) to SLB, focusing on noise with finite variance (namely, HvtLB with $\varepsilon = 1$). They proposed the AdaOFUL algorithm, achieving an optimal variance-aware regret of $\widetilde{\mathcal{O}}(d\sqrt{\sum_{t=1}^{T} \nu_t^2})$, where $\nu_t$ is the upper bound of the variance. This result is subsequently improved by Huang et al. (2023), which is also the most related work to ours. Huang et al. (2023) considered general heavy-tailed scenarios with $\varepsilon \in (0, 1]$ and proposed HEAVY-OFUL with an instance-dependent regret of $\widetilde{\mathcal{O}}\big(dT^{\frac{1-\varepsilon}{2(1+\varepsilon)}}\sqrt{\sum_{t=1}^{T} \nu_t^2} + dT^{\frac{1-\varepsilon}{2(1+\varepsilon)}}\big)$ where $\nu_t^{1+\varepsilon}$ is the $(1 + \varepsilon)$-th central moment of reward at round $t$. Subsequent works have applied Huber regression to various bandit settings, such as matrix-valued contextual bandits with low-rank structure (Kang et al., 2024). We emphasize that all the above Huber regression-based algorithms for heavy-tailed bandits are not updated in a one-pass manner. Beyond Huber-based approaches, other techniques have been explored for handling heavy-tailed noise and outliers. These include general robust loss functions like Tilted Empirical Risk Minimization (TERM) (Li et al., 2021a), and reweighting-based methods proposed in the offline contextual bandit (Sakhi et al., 2024; Behnamnia et al., 2024). A potential future direction could be to develop one-pass algorithms based on these techniques.

**One-Pass Bandit Learning.** In SLB, Abbasi-Yadkori et al. (2011) proposed the OFUL algorithm, achieving an $\mathcal{O}(d\sqrt{T})$ regret bound. It uses Least Squares (LS) for parameter estimation, naturally supporting one-pass updates with a closed-form solution. As an important extension of SLB, Generalized Linear Bandits (GLB) is first introduced by Filippi et al. (2010). Their proposed GLM-UCB algorithm achieves an $\mathcal{O}(\kappa d\sqrt{T})$ regret bound, where $\kappa$ measures the non-linearity of the generalized linear model. This algorithm relies on Maximum Likelihood Estimation (MLE) which does not have a closed-form solution like LS in SLB and requires storing all historical data, resulting in a per-round computational cost of $\mathcal{O}(t)$. To this end, Zhang et al. (2016) proposed OL2M, the first one-pass algorithm for Logistic Bandits (a specific class of GLB), achieving an $\mathcal{O}(\kappa d\sqrt{T})$ regret bound with per-round computational cost of $\mathcal{O}(1)$. Later, Jun et al. (2017) explored one-pass algorithms for GLB by introducing the online-to-confidence-set conversion. Using this conversion, they developed GLOC, a one-pass algorithm that achieves the same theoretical guarantees and per-round cost as OL2M but is applicable to all GLB. Xue et al. (2023) considered heavy-tailed GLB and proposed two one-pass algorithms: truncation-based CRTM and mean-of-medians-based CRMM, both achieving an $\widetilde{\mathcal{O}}\big(\kappa dT^{\frac{1}{1+\varepsilon}}\big)$ regret bound. Subsequent works have

tackled more challenging topics, such as reducing the dependence on $\kappa$ with one-pass algorithms or designing such algorithms for more complex models like Multinomial Logistic Bandits and Multinomial Logit MDPs (Zhang & Sugiyama, 2023; Li et al., 2024; Lee & Oh, 2024; Li et al., 2025).

## 3. One-Pass Heavy-tailed Linear Bandits

In this section, we present our one-pass method for handling linear bandits with heavy-tailed noise. Section 3.1 introduces the problem setup. Section 3.2 reviews the latest work of Huang et al. (2023) and identifies the inefficiency of the previous method. Section 3.3 introduces the OMD-type estimator based on Huber loss, along with the UCB-based arm selection strategy and the regret guarantee.

### 3.1. Problem Setup

We investigate heavy-tailed linear bandits. At round $t$, the learner chooses an arm $X_t$ from the feasible set $\mathcal{X}_t \subseteq \mathbb{R}^d$, and then the environment reveals a reward $r_t \in \mathbb{R}$ such that

$$r_t = X_t^\top \theta_* + \eta_t, \quad (4)$$

where $\theta_* \in \mathbb{R}^d$ is the unknown parameter and $\eta_t \in \mathbb{R}$ is the random noise. We define the filtration $\{\mathcal{F}_t\}_{t \geq 1}$ as $\mathcal{F}_t \triangleq \sigma(\{X_1, r_1, ..., X_{t-1}, r_{t-1}, X_t\})$. The noise $\eta_t \in \mathbb{R}$ is $\mathcal{F}_t$-measurable and satisfies for some $\varepsilon \in (0, 1]$, $\mathbb{E}[\eta_t \mid \mathcal{F}_{t-1}] = 0$ and $\mathbb{E}[|\eta_t|^{1+\varepsilon} \mid \mathcal{F}_{t-1}] = \nu_t^{1+\varepsilon}$ with $\nu_t$ being $\mathcal{F}_{t-1}$-measurable. The goal of the learner is to minimize the following (pseudo) regret,

$$\text{REG}_T = \sum_{t=1}^T \max_{\mathbf{x} \in \mathcal{X}_t} \mathbf{x}^\top \theta_* - \sum_{t=1}^T X_t^\top \theta_*.$$

We work under the following assumption as prior works (Xue et al., 2023; Huang et al., 2023).

**Assumption 1** (Boundedness). The feasible set and unknown parameter are assumed to be bounded: $\forall \mathbf{x} \in \mathcal{X}_t, \|\mathbf{x}\|_2 \leq L, \theta_* \in \Theta$ holds where $\Theta \triangleq \{\theta \mid \|\theta\|_2 \leq S\}$.

### 3.2. Reviewing Previous Efforts

In this part, we review the best-known algorithm of Huang et al. (2023). They leveraged the adaptive Huber regression method (Sun et al., 2020) to tackle the challenges of heavy-tailed noise. Specifically, adaptive Huber regression is based on the following Huber loss (Huber, 1964).

**Definition 1** (Huber loss). *Huber loss is defined as*

$$f_\tau(x) = \begin{cases} \frac{x^2}{2} & \text{if } |x| \leq \tau, \\ \tau|x| - \frac{\tau^2}{2} & \text{if } |x| > \tau, \end{cases} \quad (5)$$

*where $\tau > 0$ is the robustification parameter.*

The Huber loss is a robust modification of the squared loss that preserves convexity. It behaves as a quadratic function when $|x|$ is less than the threshold $\tau$, ensuring strong convexity in this range. When $|x|$ exceeds $\tau$, the loss transitions to a linear function to reduce the influence of outliers.

At round $t$, building on the Huber loss, adaptive Huber regression estimates the unknown parameter $\theta_*$ of linear model (4) by using all past data up to $t$ as

$$\widehat{\theta}_t = \arg\min_{\theta \in \Theta} \left\{ \frac{\lambda}{2} \|\theta\|_2^2 + \sum_{s=1}^t f_{\tau_s}\left(\frac{r_s - X_s^\top \theta}{\sigma_s}\right) \right\}, \quad (6)$$

where $\lambda$ is the regularization parameter, $f_{\tau_s}(\cdot)$ represents the Huber loss with dynamic robustification parameter $\tau_s$, and $\sigma_s$ denotes the normalization factor. Based on the adaptive Huber regression, their proposed Heavy-OFUL algorithm achieves the optimal and instance-dependent regret bound.

**Remark 1.** Note that (Kang & Kim, 2023) and (Kang et al., 2024) also use Huber regression as the estimator in different bandit problems. Adaptive Huber regression differs from their methods in two key aspects: first, it uses a time-varying robustification parameter $\tau_s$; second, it introduces a normalization factor $\sigma_s$ to further control the loss. This normalization factor not only enables instance-dependent regret bounds but also ensures that the denoised data remains within the quadratic region of the Huber loss, such that the Hessian of the loss function is guaranteed to have a lower bound, which is essential for theoretical analysis. The two prior works based on Huber regression avoid such normalization but introduce other algorithmic components and assumptions to achieve a similar effect: (i) Kang & Kim (2023) incorporates a forced exploration mechanism along with a lower-bounded Hessian assumption; (ii) Kang et al. (2024) employs a local adaptive majorize-minimization technique together with a lower bounded covariance matrix assumption to guarantee a lower bounded Hessian.

**Efficiency Concern.** When the robustification parameter $\tau_s = +\infty$, Huber regression (6) reduces to least square, allowing one-pass updates with a closed-form solution. However, for finite $\tau_s$, which is essential for handling heavy-tailed noise, (6) requires solving a strongly convex and smooth optimization problem, which needs to store all past data resulting in a storage complexity of $\mathcal{O}(t)$. Using Gradient Descent (GD) to achieve an $\epsilon$-accurate solution requires $\mathcal{O}(\log(1/\epsilon))$ iterations, with per-iteration computational cost of $\mathcal{O}(t)$ to compute the gradient over all past data. To ensure an optimal regret, $\epsilon$ is typically set to $1/\sqrt{T}$, leading to a total computational cost of $\mathcal{O}(t \log T)$ at round $t$. This would be infeasible for large-scale time horizon $T$.

### 3.3. Hvt-UCB: Huber loss-based One-Pass Algorithm

The adaptive Huber regression method proposed by Huang et al. (2023) incurs significant computational burden in each round and requires storing all past data, which is less favorable in practice. Thus, we aim to design a one-pass algorithm for HvtLB based on the Huber loss (5).

For simplicity, let $z_t(\theta) \triangleq \frac{r_t - X_t^\top \theta}{\sigma_t}$, we introduce the loss function $\ell_t(\theta)$ based on (5) as follows,

$$
\ell_t(\theta) = \begin{cases} \frac{1}{2}\left(\frac{r_t - X_t^\top \theta}{\sigma_t}\right)^2 & \text{if } |z_t(\theta)| \leq \tau_t, \\ \tau_t \left|\frac{r_t - X_t^\top \theta}{\sigma_t}\right| - \frac{\tau_t^2}{2} & \text{if } |z_t(\theta)| > \tau_t, \end{cases} \tag{7}
$$

where $\sigma_t$ is the normalization factor, and $\tau_t$ is the robustification parameter, both for controlling heavy-tailed noise.

To enable a one-pass update, instead of solving Huber regression (6) using all historical data, we adopt the Online Mirror Descent (OMD) framework (Orabona, 2019). As a general and versatile framework for online optimization, OMD has been widely applied to various challenging *adversarial* online learning problems, such as online games (Rakhlin & Sridharan, 2013; Syrgkanis et al., 2015), adversarial bandits (Abernethy et al., 2008; Wei & Luo, 2018), and dynamic regret minimization (Zhao et al., 2024; Jacobsen & Cutkosky, 2022), among others. In contrast, we leverage its potential to address *stochastic* bandit problems, drawing inspiration from recent advancements in the study of logistic bandits (Zhang & Sugiyama, 2023). Specifically, we propose the following OMD-based update for estimation,

$$
\widehat{\theta}_{t+1} = \arg\min_{\theta \in \Theta} \left\{ \langle \theta, \nabla \ell_t(\widehat{\theta}_t) \rangle + \mathcal{D}_{\psi_t}(\theta, \widehat{\theta}_t) \right\}, \tag{8}
$$

where $\Theta$ is the feasible set defined in Assumption 1 and $\mathcal{D}_{\psi_t}(\cdot, \cdot)$ is the Bregman divergence associated to the regularizer $\psi_t : \mathbb{R}^d \mapsto \mathbb{R}$. The choice of the regularizer is crucial because, in stochastic bandits, parameter estimation is subsequently used for confidence set construction. Therefore, we define the regularizer using the local information as

$$
\psi_t(\theta) = \frac{1}{2} \|\theta\|_{V_t}^2, \text{ with } V_t \triangleq \lambda I + \frac{1}{\alpha} \sum_{s=1}^{t} \frac{X_s X_s^\top}{\sigma_s^2}, \tag{9}
$$

where $V_0 \triangleq \lambda I_d$. The choice of this local-norm regularizer is compatible with the self-normalized concentration used to construct the Upper Confidence Bound (UCB) of estimation error. Here, the coefficient $\alpha$ in $V_t$ essentially represents the step size of OMD, which will be specified later.

**Computational Efficiency.** Clearly, the estimator (8) can be solved with a single projected gradient step, which can be equivalently reformulated as:

$$
\widetilde{\theta}_{t+1} = \widehat{\theta}_t - V_t^{-1} \nabla \ell_t(\widehat{\theta}_t), \quad \widehat{\theta}_{t+1} = \arg\min_{\theta \in \Theta} \left\| \theta - \widetilde{\theta}_{t+1} \right\|_{V_t},
$$

where the main computational cost lies in calculating the inverse matrix $V_t^{-1}$, which can be efficiently updated using the Sherman-Morrison-Woodbury formula with a time complexity of $\mathcal{O}(d^2)$. The projection step adds an additional time complexity of $\mathcal{O}(d^3)$. Focusing on one-pass updates and considering only the dependence on $t$ and $T$, our estimator achieves a per-round computational complexity of $\mathcal{O}(1)$, representing a significant improvement over previous methods with a complexity of $\mathcal{O}(t \log T)$. Moreover, it eliminates the need to store all historical data, requiring only $\mathcal{O}(1)$ storage cost throughout the learning process.

**Statistical Efficiency.** As mentioned earlier, OMD is a general framework that has demonstrated its effectiveness in adversarial online learning for regret minimization. Nonetheless, in stochastic bandits, a more critical requirement is the accuracy of parameter estimation. In the following, we demonstrate that OMD also performs exceptionally well in this regard. The key lemma below presents the estimation error of estimator (8) based on OMD-typed update.

**Lemma 1** (Estimation error). *If $\sigma_t, \tau_t, \tau_0$ are set as*

$$
\sigma_t = \max\left\{ \nu_t, \sigma_{\min}, \sqrt{\frac{2\beta_{t-1}}{\tau_0 \sqrt{\alpha} t^{\frac{1-\varepsilon}{2(1+\varepsilon)}}}} \|X_t\|_{V_{t-1}^{-1}} \right\},
$$

$$
\tau_t = \tau_0 \frac{\sqrt{1 + w_t^2}}{w_t} t^{\frac{1-\varepsilon}{2(1+\varepsilon)}}, \quad \tau_0 = \frac{\sqrt{2\kappa}(\log 3T)^{\frac{1-\varepsilon}{2(1+\varepsilon)}}}{\left(\log \frac{2T^2}{\delta}\right)^{\frac{1}{1+\varepsilon}}},
$$

*where $\sigma_{\min}$ is a small positive constant which will be specified later and $w_t \triangleq \frac{1}{\sqrt{\alpha}} \left\| \frac{X_t}{\sigma_t} \right\|_{V_{t-1}^{-1}}$. Let the step size $\alpha = 4$, then with probability at least $1 - 4\delta$, $\forall t \geq 1$, we have $\left\| \widehat{\theta}_{t+1} - \theta_* \right\|_{V_t} \leq \beta_t$ with*

$$
\beta_t \triangleq 107 \log \frac{2T^2}{\delta} \tau_0 t^{\frac{1-\varepsilon}{2(1+\varepsilon)}} + \sqrt{\lambda \left(2 + 4S^2\right)}, \tag{10}
$$

*where $\kappa \triangleq d \log \left(1 + \frac{L^2 T}{4\sigma_{\min}^2 \lambda d}\right)$.*

We provide an analysis sketch in Section 4, where we explain how to analyze the estimation error using the OMD framework, and decompose the effects of heavy-tailed noise and one-pass updates. The full proof of Lemma 1 is provided in Appendix B.5.

Notably, both our method and the HeavyOFUL algorithm of Huang et al. (2023) require the input of $\sigma_t, \tau_t$. In fact, our parameter setting for $\sigma_t$ is simpler. HeavyOFUL requires maintaining two complex lower bounds for $\sigma_t$ to satisfy: $\forall \theta \in \Theta, \left|\left(X_t^\top \theta - X_t^\top \theta_*\right)/\sigma_t\right| \leq \tau_t$ ensuring that the loss remains quadratic for noiseless data. In contrast, our analysis focuses on a refined condition to guarantee the quadratic loss property involving only the current estimate $\widehat{\theta}_t$: $\left|(X_t^\top \widehat{\theta}_t - X_t^\top \theta_*)/\sigma_t\right| \leq \tau_t$. This refinement enables us

**Algorithm 1** Hvt-UCB

**Input:** time horizon $T$, confidence $\delta$, regularizer $\lambda$, $\sigma_{\min}$, bounded parameters $S$, $L$, parameter for estimation $c_0$, $c_1$, $\tau_0$, $\alpha$, $\varepsilon$.

1: Set $\kappa = d \log \left(1 + \frac{L^2 T}{\sigma_{\min}^2 \lambda \alpha d}\right)$, $V_0 = \lambda I_d$, $\widehat{\theta}_1 = \mathbf{0}$ and compute $\beta_0$ by (10)
2: **for** $t = 1, 2, ..., T$ **do**
3:     Select $X_t = \arg\max_{\mathbf{x} \in \mathcal{X}_t} \langle \mathbf{x}, \widehat{\theta}_t \rangle + \beta_{t-1} \|X\|_{V_{t-1}^{-1}}$
4:     Receive the reward $r_t$, and $\nu_t$
5:     Set $\sigma_t = \max \left\{ \nu_t, \sigma_{\min}, \sqrt{\frac{2\beta_{t-1}}{\tau_0 \sqrt{\alpha} t^{\frac{1-\varepsilon}{2(1+\varepsilon)}}}} \|X_t\|_{V_{t-1}^{-1}} \right\}$
6:     Set $\tau_t = \tau_0 \frac{\sqrt{1+w_t^2}}{w_t} t^{\frac{1-\varepsilon}{2(1+\varepsilon)}}$ with $w_t = \frac{1}{\sqrt{\alpha}} \left\| \frac{X_t}{\sigma_t} \right\|_{V_{t-1}^{-1}}$
7:     Update $V_t = V_{t-1} + \alpha^{-1} \sigma_t^{-2} X_t X_t^\top$
8:     Compute $\widehat{\theta}_{t+1}$ by (8) and $\beta_t$ by (10)
9: **end for**

to design a recursive parameter setting that directly incorporates the upper confidence bound $\beta_{t-1}$ from the previous round into the configuration of $\sigma_t$ for the current round. Furthermore, this parameter setting can be directly applied to the adaptive Huber regression. Similar to Huang et al. (2023), our method requires the knowledge of the moment $\nu_t$ of each round $t$. If $\nu_t$ is not available but a general bound $\nu$ is known, such that $\mathbb{E}[|\eta_t|^{1+\varepsilon} \mid \mathcal{F}_{t-1}] \leq \nu^{1+\varepsilon}$, we can still trivially achieve the same estimation error bound by replacing $\nu_t$ with $\nu$ in the setting of $\sigma_t$.

**UCB Construction.** Based on the one-pass estimator (8) and Lemma 1, we can specify the arm selection criterion as

$$X_t = \arg\max_{\mathbf{x} \in \mathcal{X}_t} \left\{ \langle \mathbf{x}, \widehat{\theta}_t \rangle + \beta_{t-1} \|\mathbf{x}\|_{V_{t-1}^{-1}} \right\}. \quad (11)$$

Algorithm 1 summarizes the overall algorithm, and enjoys following instance-dependent regret.

**Theorem 1.** *By setting $\sigma_t$, $\tau_t$, $\tau_0$, $\alpha$ as in Lemma 1, and let $\lambda = d$, $\sigma_{\min} = \frac{1}{\sqrt{T}}$, $\delta = \frac{1}{8T}$, with probability at least $1 - 1/T$, the regret of Hvt-UCB is bounded by*

$$\mathrm{REG}_T \leq \widetilde{\mathcal{O}} \left( d T^{\frac{1-\varepsilon}{2(1+\varepsilon)}} \sqrt{\sum_{t=1}^{T} \nu_t^2} + d T^{\frac{1-\varepsilon}{2(1+\varepsilon)}} \right).$$

The proof of Theorem 1 can be found in Appendix C.1. Compared to Huang et al. (2023), our approach achieves the same order of instance-dependent regret bounds, while significantly reducing the per-round computational complexity from $\mathcal{O}(t \log T)$ to $\mathcal{O}(1)$.

When the moment $\nu_t$ of each round $t$ is unknown, but a general bound $\nu$ is known, such that $\mathbb{E}[|\eta_t|^{1+\varepsilon} \mid \mathcal{F}_{t-1}] \leq \nu^{1+\varepsilon}$, we can achieve the following regret bound.

**Corollary 1.** *Follow the parameter setting in Theorem 1, and replace the $\nu_t$ with $\nu$ in the $\sigma_t$ setting, the regret of Hvt-UCB is bounded with probability at least $1 - 1/T$, by*

$$\mathrm{REG}_T \leq \widetilde{\mathcal{O}} \left( d T^{\frac{1}{1+\varepsilon}} \right).$$

The proof of Corollary 1 can be found in Appendix C.2. This bound matches the lower bound $\Omega(d T^{\frac{1}{1+\varepsilon}})$ (Shao et al., 2018) up to logarithmic factors. Moreover, when $\varepsilon = 1$, i.e., the bounded variance setting, our approach recovers the regret bound $\widetilde{\mathcal{O}}(d\sqrt{T})$, which nearly matches the lower bound $\Omega(d\sqrt{T})$ (Dani et al., 2008).

## 4. Analysis Sketch

In this section, we provide a proof sketch for Lemma 1, where we adapt the analysis framework of Online Mirror Descent (OMD) to the SLB setting. To simplify the presentation, we first illustrate this framework using a simpler case where the noise follows a sub-Gaussian distribution in Section 4.1. We then discuss the challenges of extending this framework to handle heavy-tailed noise and present our refined analysis in Section 4.2.

### 4.1. Case study of sub-Gaussian noise

We first consider that the noise $\eta_t$ in model (4) follows a $R$-sub-Gaussian distribution. Then the robustification parameter $\tau_t$ should be set as $+\infty$ and the Huber loss recovers the squared loss by $\ell_t(\theta) = \frac{1}{2}((r_t - X_t^\top \theta)/\sigma_t)^2$.

**Estimation error decomposition.** We begin by decomposing the estimation error into the following three components:

**Lemma 2.** *Let $\widetilde{\ell}_t(\theta) = \frac{1}{2}(X_t^\top(\theta_* - \theta)/\sigma_t)^2$ be the denoised Huber loss function, then the estimation error of estimator (8) with $\alpha = 2$ can be decomposed as follows,*

$$\left\|\widehat{\theta}_{t+1} - \theta_*\right\|_{V_t}^2 \leq 2 \underbrace{\sum_{s=1}^{t} \left\langle \nabla \widetilde{\ell}_s(\widehat{\theta}_s) - \nabla \ell_s(\widehat{\theta}_s), \widehat{\theta}_s - \theta_* \right\rangle}_{\text{generalization gap term}}$$

$$+ \underbrace{\sum_{s=1}^{t} \left\|\nabla \ell_s(\widehat{\theta}_s)\right\|_{V_s^{-1}}^2}_{\text{stability term}} - \frac{1}{2} \underbrace{\sum_{s=1}^{t} \left\|\widehat{\theta}_s - \theta_*\right\|_{\frac{X_s X_s^\top}{\sigma_s^2}}^2}_{\text{negative term}} + 4\lambda S^2,$$

*Proof Sketch.* First based on the standard analysis of OMD (Orabona, 2019, Lemma 6.16), we have

$$\left\langle \nabla \ell_t(\widehat{\theta}_t), \widehat{\theta}_t - \theta_* \right\rangle$$
$$\leq \frac{1}{2} \left\|\widehat{\theta}_t - \theta_*\right\|_{V_t}^2 - \frac{1}{2} \left\|\widehat{\theta}_{t+1} - \theta_*\right\|_{V_t}^2 + \frac{1}{2} \left\|\nabla \ell_t(\widehat{\theta}_t)\right\|_{V_t^{-1}}^2.$$

Based on the quadratic property of the denoised loss function $\widetilde{\ell}_t(\cdot)$, we have the following equality,

$$\widetilde{\ell}_t(\widehat{\theta}_t) - \widetilde{\ell}_t(\theta_*) = \left\langle \nabla\widetilde{\ell}_t(\widehat{\theta}_t), \widehat{\theta}_t - \theta_* \right\rangle - \frac{1}{2}\left\|\widehat{\theta}_t - \theta_*\right\|^2_{\frac{X_t X_t^\top}{\sigma_t^2}}.$$

Since $\theta_*$ is the minimizer of $\widetilde{\ell}_t(\cdot)$, $\widetilde{\ell}_t(\widehat{\theta}_t) - \widetilde{\ell}_t(\theta_*) \geq 0$, which implies that $\frac{1}{2}\left\|\widehat{\theta}_t - \theta_*\right\|^2_{X_t X_t^\top/\sigma_t^2} \leq \langle\nabla\widetilde{\ell}_t(\widehat{\theta}_t), \widehat{\theta}_t - \theta_*\rangle$, then substituting this into the OMD update rule and we can achieve the iteration inequality for the estimation error, further with a telescoping sum, and we finish the proof. $\square$

**Stability term analysis.** The first step is to extract stochastic term and deterministic term at each step $s$ as follows,

$$\left\|\nabla\ell_s(\widehat{\theta}_s)\right\|^2_{V_s^{-1}} = \left(\frac{X_s^\top(\theta_* - \widehat{\theta}_s)}{\sigma_s} + \frac{\eta_s}{\sigma_s}\right)^2 \left\|\frac{X_s}{\sigma_s}\right\|^2_{V_s^{-1}}.$$

Setting $\sigma_t = 1$, we have $\left|X_s^\top(\theta_* - \widehat{\theta}_s)\right| \leq 2LS$. Then based on the sub-Gaussian property, we have $|\eta_s|$ bounded by $\mathcal{O}(\sqrt{\log T})$ with high probability, and $\sum_{s=1}^t \|X_s\|^2_{V_s^{-1}}$ can be bounded by $\mathcal{O}(\log T)$ with Lemma 7 (potential lemma), which means $\sum_{s=1}^t \left\|\nabla\ell_s(\widehat{\theta}_s)\right\|^2_{V_s^{-1}} \leq \mathcal{O}(\log^2 T)$.

**Generalization gap term analysis.** By taking the gradient of the loss function, we have

$$\sum_{s=1}^t \left\langle \nabla\widetilde{\ell}_s(\widehat{\theta}_s) - \nabla\ell_s(\widehat{\theta}_s), \widehat{\theta}_s - \theta_* \right\rangle = \sum_{s=1}^t \eta_s \left\langle X_s, \widehat{\theta}_s - \theta_* \right\rangle,$$

then we can directly apply the 1-dimension self-normalized concentration (Theorem 2), further based on AM-GM inequality, we can bound the general gap term by a $\mathcal{O}(\log T)$ term and $\frac{1}{4}\sum_{s=1}^t \left\|\widehat{\theta}_s - \theta_*\right\|^2_{X_s X_s^\top}$, which can be further canceled by the negative term.

By integrating the stability term and generalization gap term into Lemma 2, we can achieve the upper bound for the estimation error as $\left\|\widehat{\theta}_{t+1} - \theta_*\right\|^2_{V_t} \leq \mathcal{O}(\log T)$.

### 4.2. Analysis for heavy-tailed noise

In this section, we extend our analysis to the heavy-tailed case. The presence of heavy-tailed noise and the partially linear structure of the Huber loss introduces new challenges in the three key analyses mentioned above. We will address these challenges one by one in the following discussion.

**Estimation error decomposition.** With the presence of heavy-tailed noise, the denoised Huber loss function with noise-free data $\widetilde{z}_t(\theta) \triangleq \frac{X_t^\top\theta_* - X_t^\top\theta}{\sigma_t}$ is defined as

$$\widetilde{\ell}_t(\theta) \triangleq \begin{cases} \frac{1}{2}\left(\frac{X_t^\top(\theta_* - \theta)}{\sigma_t}\right)^2 & \text{if } |\widetilde{z}_t(\theta)| \leq \tau_t, \\ \tau_t\left|\frac{X_t^\top(\theta_* - \theta)}{\sigma_t}\right| - \frac{\tau_t^2}{2} & \text{if } |\widetilde{z}_t(\theta)| > \tau_t. \end{cases}$$

Now $\widetilde{\ell}_t(\theta)$ is partially quadratic and partially linear, to further utilize the OMD-based estimation error decomposition, we need to make sure $\widehat{\theta}_t$ and $\theta_*$ both lie in the quadratic region of the loss function $\widetilde{\ell}_t(\cdot)$. Assuming the event $A_t = \left\{\forall s \in [t], \left\|\widehat{\theta}_s - \theta_*\right\|_{V_{s-1}} \leq \beta_{s-1}\right\}$ holds, we set $\sigma_t^2 \geq (2\|X_t\|^2_{V_{t-1}^{-1}}\beta_{t-1})/(\sqrt{\alpha}\tau_0 t^{\frac{1-\varepsilon}{2(1+\varepsilon)}})$, we can ensure $\left|(X_t^\top\widehat{\theta}_t - X_t^\top\theta_*)/\sigma_t\right| \leq \frac{\tau_t}{2}$. This guarantees that $\widetilde{\ell}_t(\widehat{\theta}_t)$ is always in the quadratic region, allowing us to continue using the previous decomposition. The negative term from the decomposition will then be used for subsequent cancellation.

**Stability term analysis.** Similar to the sub-Gaussian case, we can decompose the stability term into two stochastic and deterministic terms as follows,

$$\sum_{s=1}^t \left\|\nabla\ell_s(\widehat{\theta}_s)\right\|^2_{V_s^{-1}} \leq 2\underbrace{\sum_{s=1}^t \left(\min\left\{\left|\frac{\eta_s}{\sigma_s}\right|, \tau_s\right\}\right)^2 \left\|\frac{X_s}{\sigma_s}\right\|^2_{V_s^{-1}}}_{\text{stochastic term}}$$
$$+ 2\underbrace{\sum_{s=1}^t \left(\frac{X_s^\top\theta_* - X_s^\top\widehat{\theta}_s}{\sigma_s}\right)^2 \left\|\frac{X_s}{\sigma_s}\right\|^2_{V_s^{-1}}}_{\text{deterministic term}}.$$

Note that the stochastic term is no longer inherently bounded. To address this, we utilize a concentration technique (Lemma 6) to bound it by $\widetilde{\mathcal{O}}(t^{\frac{1-\varepsilon}{1+\varepsilon}})$. For the deterministic term, $\frac{X_s^\top\theta_* - X_s^\top\widehat{\theta}_s}{\sigma_s}$ is bounded by $2LS/\sigma_{\min}$, where $\sigma_{\min}$ is the minimum of $\sigma_t$, which can be as small as $1/\sqrt{T}$. If we continue using the analysis based on potential lemma, this term will grow to $\widetilde{\mathcal{O}}(\sqrt{T})$, which is undesirable. Based on the settings of $\sigma_t$, we ensure that $\left\|\frac{X_s}{\sigma_s}\right\|^2_{V_s^{-1}} \leq \frac{1}{8}$. Therefore, this term can be bounded by $\frac{1}{4}\sum_{s=1}^t \left\|\theta_* - \widehat{\theta}_s\right\|^2_{X_s X_s^\top/\sigma_s^2}$, which can be further canceled out by the negative term.

**Generalization gap term analysis.** The non-linear gradient of the Huber loss in the heavy-tailed setting makes it impossible to directly obtain a one-dimensional self-normalized term. We first decompose it into two distinct components as follows,

$$2\underbrace{\sum_{s=1}^t \left\langle \nabla\widetilde{\ell}_s(\widehat{\theta}_s) + \nabla\ell_s(\theta_*) - \nabla\ell_s(\widehat{\theta}_s), \widehat{\theta}_s - \theta_* \right\rangle}_{\text{Huber-loss term}}$$
$$+ 2\underbrace{\sum_{s=1}^t \left\langle -\nabla\ell_s(\theta_*), \widehat{\theta}_s - \theta_* \right\rangle}_{\text{self-normalized term}},$$

where Huber-loss term represents the influence of the Huber loss structure on the estimation error. Specifically, when all three gradient terms are within the quadratic region of

the Huber loss, i.e., when $|z_t(\theta_*)| \leq \frac{\tau_t}{2}$, this term becomes zero. Consequently, this term reduces to a sum involving the indicator function $\sum_{s=1}^{t} \mathbb{1}\left\{|z_s(\theta_*)| > \frac{\tau_s}{2}\right\}$, which can be bounded by $\widetilde{\mathcal{O}}\left(t^{\frac{1-\varepsilon}{2(1+\varepsilon)}}\beta_t\right)$ using Eq. (C.12) of Huang et al. (2023). Then the self-normalized term corresponds to the 1-dimensional version of the self-normalized concentration can be bounded by an $\widetilde{\mathcal{O}}(t^{\frac{1-\varepsilon}{1+\varepsilon}})$ term and $\frac{1}{2}\sum_{s=1}^{t}\|\widehat{\theta}_s - \theta_*\|^2_{X_s X_s^\top / \sigma_s^2}$ will be canceled by the negative term.

By combining the analysis for stability term and generalization gap term and substituting back to the estimation error decomposition, we obtain:

$$\left\|\widehat{\theta}_{t+1} - \theta_*\right\|^2_{V_t} \lesssim t^{\frac{1-\varepsilon}{2(1+\varepsilon)}}\beta_t + t^{\frac{1-\varepsilon}{1+\varepsilon}}.$$

To ensure the event $A_t$ holds, $\beta_t$ must satisfy:

$$\beta_t^2 \gtrsim t^{\frac{1-\varepsilon}{2(1+\varepsilon)}}\beta_t + t^{\frac{1-\varepsilon}{1+\varepsilon}}.$$

Finally, solving for $\beta_t$, we get $\beta_t = \mathcal{O}(t^{\frac{1-\varepsilon}{2(1+\varepsilon)}})$. Thus we finish the proof of Lemma 1.

## 5. Experiments

In this section, we evaluate the empirical performance and time efficiency of our proposed Hvt-UCB algorithm. We present two experiments under heavy-tailed noise (Student's $t$) and light-tailed noise (Gaussian). Furthermore, we provide additional experiments in Appendix E, including various heavy-tailed noises (Pareto, Lomax, Fisk), time-varying $\nu_t$, and changing arm sets $\mathcal{X}_t$, to further validate the effectiveness and robustness of our algorithm.

**Settings.** We consider the linear model $r_t = X_t^\top \theta_* + \eta_t$, where the dimension $d = 2$, the number of rounds $T = 18000$, and the number of arms $n = 50$. Each dimension of the feature vectors for the arms is uniformly sampled from $[-1, 1]$ and subsequently rescaled to satisfy $L = 1$. Similarly, $\theta_*$ is sampled in the same way and rescaled to satisfy $S = 1$. We conduct two synthetic experiments with different distributions for noise $\eta_t$: (a) Student's $t$-distribution with degree of freedom $df = 2.1$ to represent heavy-tailed noise; and (b) Gaussian noise sampled from $\mathcal{N}(0, 1)$ to represent light-tailed noise. For the heavy-tailed experiment, we set $\varepsilon = 0.99$ and $\nu = 1.31$, while for the light-tailed experiment, we set $\varepsilon = 1$. We compare the performance of our proposed Hvt-UCB algorithm with the following baselines: (a) the OFUL algorithm (Abbasi-Yadkori et al., 2011); (b) the one-pass truncation-based algorithm CRTM (Xue et al., 2023); (c) the one-pass MOM-based algorithm CRMM (Xue et al., 2023); and (d) the Huber regression-based algorithm HEAVY-OFUL (Huang et al., 2023).

**Results.** We conducted 10 independent trials and averaged the results. Figure 1 shows the cumulative regret

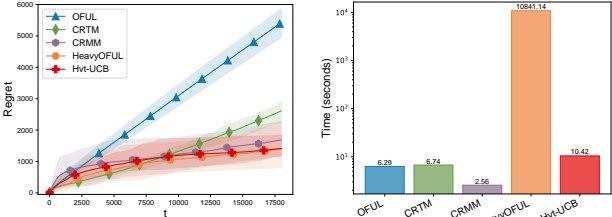

Figure 1: Student $t$: regret and time cost.

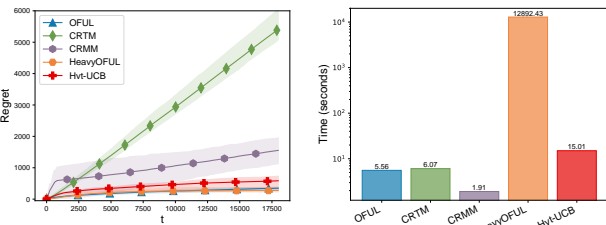

Figure 2: Gaussian: regret and time cost.

and computational time under Student's $t$ noise, while Figure 2 presents the same metrics under Gaussian noise, with shaded regions indicating the variance across trials. Under both noise settings, our algorithm demonstrates comparable regret performance to Heavy-OFUL while achieving remarkable computational efficiency, with a speedup exceeding $800\times$. This efficiency advantage becomes more important as the number of rounds $T$ increases. OFUL, CTRM, and CRMM are faster than our Hvt-UCB algorithm because they use closed-form least squares solutions, while Hvt-UCB requires a projection step in each round. However, Hvt-UCB's runtime remains competitive and is on the same order of magnitude as these least squares-based algorithms. Notably, among the one-pass algorithms, CRMM exhibits the fastest runtime because it updates its estimates periodically (every $\mathcal{O}(\log T)$ rounds) rather than in every round. In the heavy-tailed noise scenario (Figure 1), the performance of OFUL drops significantly, demonstrating its sensitivity to heavy-tailed distributions. In the Gaussian noise setting (Figure 2), our algorithm maintains competitive performance alongside OFUL. However, the truncation-based CRTM algorithm shows suboptimal performance due to excessive reward truncation in light-tailed noise scenarios, which leads to the loss of valuable data. These experimental results validate that our Hvt-UCB algorithm achieves robust performance across heavy-tailed and light-tailed noise environments while maintaining low computational cost.

## 6. Discussion

In this section, we discuss the potential applications of our approach and highlight its advantages compared to previous methods in broader scenarios involving heavy-tailed noise. Specifically, we focus on two settings: heavy-tailed linear MDPs and adaptive control under heavy-tailed noise.

## 6.1. Online Linear MDPs

In the setting of heavy-tailed linear MDPs (Li & Sun, 2024; Huang et al., 2023), the realizable reward for taking action $a$ in state $s$ at the $h$-th episode is given by:

$$R_h(s, a) = \langle \phi(s, a), \boldsymbol{\theta}_h^* \rangle + \epsilon_h(s, a),$$

where $\phi(s, a) \in \mathbb{R}^d$ is known feature map, $\boldsymbol{\theta}_h^* \in \mathbb{R}^d$ is the unknown parameter, and $\epsilon_h(s, a)$ is the heavy-tailed noise with $\mathbb{E}_h[\epsilon_h](s, a) = 0$ and $\mathbb{E}_h[|\epsilon_h|^{1+\varepsilon}](s, a) \leq \nu_R^{1+\varepsilon}$.

Linear MDPs often adopt techniques from linear bandits, including estimating unknown parameters in a linear model and constructing an upper confidence bound (UCB) based on estimation error analysis. For heavy-tailed linear MDPs, (Huang et al., 2023) employs adaptive Huber regression to estimate the unknown parameter $\boldsymbol{\theta}_h^*$ in the reward function. This introduces a challenge similar to that in HvtLB: adaptive Huber regression is an offline algorithm, while an online setting requires a one-pass estimator. Existing one-pass estimators for HvtLB, such as the truncation-based and MOM-based algorithms proposed by Xue et al. (2023), are not applicable to heavy-tailed linear MDPs. Truncation-based algorithms rely on assumptions about absolute moments, which are not suitable in this case. MOM-based methods require repeated reward observations for a fixed $\phi(s, a)$, which is infeasible in linear MDPs since state transitions occur probabilistically.

In contrast, our proposed Hvt-UCB method does not suffer the aforementioned issue and has the potential to be applied to this scenario without requiring additional assumptions. Its estimation error bound analysis naturally extends to linear MDPs, enabling the construction of UCBs for the value function. By leveraging the UCB-to-regret analysis in (Huang et al., 2023), Hvt-UCB has the potential to achieve the same theoretical guarantees as adaptive Huber regression, while significantly reducing computational costs.

## 6.2. Online Adaptive Control

The online adaptive control of Linear Quadratic Systems (Abbasi-Yadkori & Szepesvári, 2011) considers the following state transition system,

$$x_{t+1} = A x_t + B u_t + w_{t+1},$$

where $x_t \in \mathbb{R}^n$ represents the state at time $t$, $u_t \in \mathbb{R}^d$ is the control input at time $t$, $A \in \mathbb{R}^{n \times n}$ and $B \in \mathbb{R}^{n \times d}$ are unknown system matrices, and $w_{t+1}$ is the noise. The online adaptive control requires estimating the unknown parameters $A$ and $B$ (also known as system identification), and constructing finite-sample guarantees for estimation error. To this end, Abbasi-Yadkori & Szepesvári (2011) transformed the system into the following linear model:

$$x_{t+1} = \Theta_*^\top z_t + w_{t+1},$$

where $\Theta_* = [A; B]$, $z_t = [x_t; u_t]$ and the noise $w_t$ in each dimension is assumed to be sub-Gaussian. Then, they used a least squares estimator and leveraged the self-normalized concentration technique from linear bandit analysis (Abbasi-Yadkori et al., 2011) to estimate $\Theta_*$ and provide finite-sample estimation error guarantees.

As noted by Tsiamis et al. (2023), finite-sample guarantees for system identification under heavy-tailed noise remain an open challenge. In light of Abbasi-Yadkori & Szepesvári (2011)'s application of linear bandit techniques to sub-Gaussian cases, we believe our method for HvtLB can be also benefit adaptive control in heavy-tailed settings. However, MOM methods are infeasible as the feature $z_t$, which includes the evolving state $x_t$, cannot be repeatedly sampled, and truncation-based methods rely on assumptions of bounded absolute moments for states, which may not hold. While adaptive Huber regression (Li & Sun, 2024; Huang et al., 2023) offers promising theoretical guarantees for heavy-tailed linear system identification, its computational inefficiency makes it unsuitable for adaptive control. In contrast, our proposed Hvt-UCB algorithm provides one-pass updates and has the potential to deliver finite-sample guarantees for adaptive control under heavy-tailed noise.

## 7. Conclusion

In this paper, we investigate the problem of heavy-tailed linear bandits (HvtLB). We highlight the advantages of Huber loss-based methods over truncation and median-of-means strategies, while also identifying the inefficiencies in previous Huber loss-based methods for HvtLB. To address these issues, we propose a Huber loss-based one-pass algorithm Hvt-UCB, based on Online Mirror Descent (OMD), a well-known regret minimization framework in online learning, which we adapt here for use as a statistical estimator. Hvt-UCB achieves the optimal and instance-dependent regret bound $\widetilde{\mathcal{O}}\big(dT^{\frac{1-\varepsilon}{2(1+\varepsilon)}} \sqrt{\sum_{t=1}^T \nu_t^2} + dT^{\frac{1-\varepsilon}{2(1+\varepsilon)}}\big)$ while only requiring $\mathcal{O}(1)$ per-round computational cost. Furthermore, our proposed method enjoys the potential to be extended to more broad online decision-making problems, including online linear MDPs and online adaptive control, thereby broadening its applicability. The key contribution is our adaptation of the OMD framework to stochastic linear bandits, effectively addressing the challenges introduced by heavy-tailed noise and the structure of the Huber loss.

Both our work and Huang et al. (2023) rely on prior knowledge of the moment bound $\nu_t$ to achieve instance-dependent guarantees. Recent progress in heavy-tailed multi-armed bandits has removed this requirement in different settings: adversarial (Huang et al., 2022), stochastic (Genalti et al., 2024), and best-of-both-worlds (Chen et al., 2025). Whether similar techniques can be applied to heavy-tailed linear bandits to relax this assumption remains an open question.

## Acknowledgements

This research was supported by National Science and Technology Major Project (2022ZD0114800), NSFC (U23A20382), Postgraduate Research & Practice Innovation Program of Jiangsu Province (KYCX25_0327). The authors also thank Bo Xue for helpful discussions.

## Impact Statement

This paper presents work whose goal is to advance the field of Machine Learning. There are many potential societal consequences of our work, none of which we feel must be specifically highlighted here.

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

## A. Properties of Huber Loss

**Property 1** (Property 1 of Huang et al. (2023))**.** *Let $f_\tau(x)$ be the Huber loss defined in Definition 1, then the followings are true:*

1. $|f'_\tau(x)| = \min\{|x|, \tau\}$,

2. $f'_\tau(x) = \tau f'_1\left(\frac{x}{\tau}\right)$,

3. $-\log\left(1 - x + |x|^{1+\varepsilon}\right) \le f'_1(x) \le \log\left(1 + x + |x|^{1+\varepsilon}\right)$ *for any $\varepsilon \in (0,1]$,*

4. $f''_\tau(x) = \mathbb{1}\{|x| \le \tau\}$.

Furthermore, the gradient of the loss function (7) is,

$$\nabla \ell_t(\theta) = \begin{cases} -z_t(\theta)\frac{X_t}{\sigma_t} & \text{if } |z_t(\theta)| \le \tau_t, \\ -\tau_t \frac{X_t}{\sigma_t} & \text{if } z_t(\theta) > \tau_t, \\ \tau_t \frac{X_t}{\sigma_t} & \text{if } z_t(\theta) < -\tau_t. \end{cases} \tag{12}$$

Based on Property 1, we have

$$\|\nabla \ell_t(\theta)\| = \left\|\min\left\{|z_t(\theta)|, \tau_t\right\}\frac{X_t}{\sigma_t}\right\|, \quad \nabla^2 \ell_t(\theta) = \mathbb{1}\left\{|z_t(\theta)| \le \tau_t\right\}\frac{X_t X_t^\top}{\sigma_t^2}. \tag{13}$$

## B. Estimation Error Analysis

For the analysis of estimation error, we begin by defining the following denoised loss function based on noise-free data $\widetilde{z}_t(\theta) \triangleq \frac{X_t^\top \theta_* - X_t^\top \theta}{\sigma_t}$,

$$\widetilde{\ell}_t(\theta) \triangleq \begin{cases} \frac{1}{2}\left(\frac{X_t^\top(\theta_* - \theta)}{\sigma_t}\right)^2 & \text{if } |\widetilde{z}_t(\theta)| \le \tau_t, \\ \tau_t\left|\frac{X_t^\top(\theta_* - \theta)}{\sigma_t}\right| - \frac{\tau_t^2}{2} & \text{if } |\widetilde{z}_t(\theta)| > \tau_t. \end{cases}$$

The gradient and Hessian of the denoised loss function $\widetilde{\ell}_t(\theta)$ can be writen as

$$\nabla \widetilde{\ell}_t(\theta) = \begin{cases} -\widetilde{z}_t(\theta)\frac{X_t}{\sigma_t} & \text{if } |\widetilde{z}_t(\theta)| \le \tau_t, \\ -\tau_t \frac{X_t}{\sigma_t} & \text{if } \widetilde{z}_t(\theta) > \tau_t, \\ \tau_t \frac{X_t}{\sigma_t} & \text{if } \widetilde{z}_t(\theta) < -\tau_t. \end{cases} \qquad \nabla^2 \widetilde{\ell}_t(\theta) = \mathbb{1}\left\{|\widetilde{z}_t(\theta)| \le \tau_t\right\}\frac{X_t X_t^\top}{\sigma_t^2}. \tag{14}$$

We further set the robustification parameter $\tau_t$ as following,

$$\tau_t = \tau_0 \frac{\sqrt{1 + w_t^2}}{w_t} t^{\frac{1-\varepsilon}{2(1+\varepsilon)}}, \quad \tau_0 = \frac{\sqrt{2\kappa}(\log 3T)^{\frac{1-\varepsilon}{2(1+\varepsilon)}}}{\left(\log \frac{2T^2}{\delta}\right)^{\frac{1}{1+\varepsilon}}}, \quad \text{where } w_t \triangleq \frac{1}{\sqrt{\alpha}}\left\|\frac{X_t}{\sigma_t}\right\|_{V_{t-1}^{-1}}, \tag{15}$$

and we denote event $A_t = \left\{\forall s \in [t], \left\|\widehat{\theta}_s - \theta_*\right\|_{V_{s-1}} \le \beta_{s-1}\right\}$. Based on the parameter setting and the event $A_t$, we derive three useful lemmas for the analysis of estimation error in the following section.

### B.1. Useful Lemmas

In this section, we provide some useful lemmas for the estimation error analysis. We provide the estimation error decomposition in Lemma 3, the stability term analysis in Lemma 4, and the generalization gap term analysis in Lemma 5.

**Lemma 3** (Estimation error decomposition)**.** *When event $A_t$ holds, by setting $\tau_t$ as (15), and $\sigma_t$ as*

$$\sigma_t^2 \ge \frac{2\|X_t\|_{V_{t-1}^{-1}}^2 \beta_{t-1}}{\sqrt{\alpha}\tau_0 t^{\frac{1-\varepsilon}{2(1+\varepsilon)}}}, \tag{16}$$

*then the estimation error can be decomposed as following three terms,*

$$\left\|\widehat{\theta}_{t+1} - \theta_*\right\|_{V_t}^2$$
$$\leq 4\lambda S^2 + \sum_{s=1}^{t}\left\|\nabla\ell_s(\widehat{\theta}_s)\right\|_{V_s^{-1}}^2 + 2\sum_{s=1}^{t}\left\langle\nabla\widetilde{\ell}_s(\widehat{\theta}_s) - \nabla\ell_s(\widehat{\theta}_s), \widehat{\theta}_s - \theta_*\right\rangle + \left(\frac{1}{\alpha} - 1\right)\sum_{s=1}^{t}\left\|\widehat{\theta}_s - \theta_*\right\|_{\frac{X_s X_s^\top}{\sigma_s^2}}^2.$$

**Lemma 4.** *By setting $\tau_t$ as (15) and $\sigma_t = \max\left\{\nu_t, \sigma_{\min}, 2\sqrt{2}\left\|X_t\right\|_{V_{t-1}^{-1}}\right\}$, with probability at least $1 - \delta$, we have $\forall t \geq 1$,*

$$\sum_{s=1}^{t}\left\|\nabla\ell_s(\widehat{\theta}_s)\right\|_{V_s^{-1}}^2 \leq 6\alpha\left(t^{\frac{1-\varepsilon}{2(1+\varepsilon)}}\sqrt{2\kappa}(\log 3T)^{\frac{1-\varepsilon}{2(1+\varepsilon)}}\left(\log\frac{2T^2}{\delta}\right)^{\frac{\varepsilon}{1+\varepsilon}}\right)^2 + \frac{1}{4}\sum_{s=1}^{t}\left\|\theta_* - \widehat{\theta}_s\right\|_{\frac{X_s X_s^\top}{\sigma^2}}^2.$$

*where $\alpha$ is the learning rate need to be tuned and $\kappa \triangleq d\log\left(1 + \frac{L^2 T}{\sigma_{\min}^2\lambda\alpha d}\right)$.*

**Lemma 5.** *By setting $\tau_t$ as (15) and $\sigma_t$ as following,*

$$\sigma_t = \max\left\{\nu_t, \sigma_{\min}, 2\sqrt{2}\left\|X_t\right\|_{V_{t-1}^{-1}}, \sqrt{\frac{2\left\|X_t\right\|_{V_{t-1}^{-1}}^2\beta_{t-1}}{\sqrt{\alpha}\tau_0 t^{\frac{1-\varepsilon}{2(1+\varepsilon)}}}}\right\}, \tag{17}$$

*with probability at least $1 - 3\delta$, we have $\forall t \geq 1$,*

$$\sum_{s=1}^{t}\left\langle\nabla\widetilde{\ell}_s(\widehat{\theta}_s) - \nabla\ell_s(\widehat{\theta}_s), \widehat{\theta}_s - \theta_*\right\rangle\mathbb{1}_{A_s} \leq 23\sqrt{\alpha + \frac{1}{8}}\log\frac{2T^2}{\delta}\tau_0 t^{\frac{1-\varepsilon}{2(1+\varepsilon)}}\max_{s\in[t+1]}\beta_{s-1}$$

$$+ \frac{1}{4}\left(\lambda\alpha + \sum_{s=1}^{t}\left\langle\frac{X_s}{\sigma_s}, \widehat{\theta}_s - \theta_*\right\rangle^2\right) + \left(8t^{\frac{1-\varepsilon}{2(1+\varepsilon)}}\sqrt{2\kappa}(\log 3T)^{\frac{1-\varepsilon}{2(1+\varepsilon)}}\left(\log\left(\frac{2T^2}{\delta}\right)\right)^{\frac{\varepsilon}{1+\varepsilon}}\right)^2.$$

## B.2. Proof of Lemma 3

*Proof.* Since $A_t = \left\{\forall s\in[t], \left\|\widehat{\theta}_s - \theta_*\right\|_{V_{s-1}} \leq \beta_{s-1}\right\}$ holds, $\tau_t$ and $\sigma_t$ satisfies (15) and (16), we have

$$\left|\frac{X_t^\top\widehat{\theta}_t - X_t^\top\theta_*}{\sigma_t}\right| \leq \left\|\frac{X_t}{\sigma_t}\right\|_{V_{t-1}^{-1}}\left\|\theta_* - \widehat{\theta}_t\right\|_{V_{t-1}} \leq \frac{\tau_0 t^{\frac{1-\varepsilon}{2(1+\varepsilon)}}}{2w_t\beta_{t-1}}\beta_{t-1} \leq \frac{1}{2}\tau_0\frac{\sqrt{1+w_t^2}}{w_t}t^{\frac{1-\varepsilon}{2(1+\varepsilon)}} = \frac{\tau_t}{2}, \tag{18}$$

where $w_t = \frac{1}{\sqrt{\alpha}}\left\|\frac{X_t}{\sigma_t}\right\|_{V_{t-1}^{-1}}$. Then, based on definition of $\widetilde{z}_t(\cdot)$, we have $\widetilde{z}_t(\theta_*) = 0 \leq \tau_t$, and (18) shows that $\widetilde{z}_t(\widehat{\theta}_t) \leq \frac{\tau_t}{2} \leq \tau_t$, which means both points $\theta_*$ and $\widehat{\theta}_t$ lie on the quadratic side of the denoised loss function $\widetilde{\ell}_t(\cdot)$. This allows us to apply Taylor's Formula with lagrange remainder to obtain

$$\widetilde{\ell}_t(\theta_*) = \widetilde{\ell}_t(\widehat{\theta}_t) + \left\langle\nabla\widetilde{\ell}_t(\widehat{\theta}_t), \theta_* - \widehat{\theta}_t\right\rangle + \frac{1}{2}\left\|\widehat{\theta}_t - \theta_*\right\|_{\nabla^2\widetilde{\ell}_t(\xi_t)}^2, \tag{19}$$

where $\xi_t = \gamma\widehat{\theta}_t + (1-\gamma)\theta_*$ for some $\gamma \in (0,1)$, which means $\xi_t$ also lie on the quadratic side of the denoised loss function:

$$|\widetilde{z}_t(\xi_t)| = \left|\frac{X_t^\top\xi_t - X_t^\top\theta_*}{\sigma_t}\right| = \left|\frac{X_t^\top(\gamma\widehat{\theta}_t + (1-\gamma)\theta_*) - X_t^\top\theta_*}{\sigma_t}\right| = \gamma\left|\frac{X_t^\top\widehat{\theta}_t - X_t^\top\theta_*}{\sigma_t}\right| \leq \gamma\frac{\tau_t}{2} \leq \tau_t,$$

where the first inequality comes from (18). Then we have $\nabla^2\widetilde{\ell}_t(\xi_t) = \mathbb{1}\left\{\widetilde{z}_t(\xi_t) \leq \tau_t\right\}\frac{X_t X_t^\top}{\sigma_t^2} = \frac{X_t X_t^\top}{\sigma_t^2}$. At the same time, since $\theta_* = \arg\min_{\theta\in\mathbb{R}^d}\widetilde{\ell}_t(\theta)$, we have $0 \leq \widetilde{\ell}_t(\widehat{\theta}_t) - \widetilde{\ell}_t(\theta_*)$. Substituting these into (19), we have

$$0 \leq \widetilde{\ell}_t(\widehat{\theta}_t) - \widetilde{\ell}_t(\theta_*) = \left\langle\nabla\widetilde{\ell}_t(\widehat{\theta}_t), \widehat{\theta}_t - \theta_*\right\rangle - \frac{1}{2}\left\|\widehat{\theta}_t - \theta_*\right\|_{\frac{X_t X_t^\top}{\sigma_t^2}}^2, \tag{20}$$

which means $\frac{1}{2}\left\|\widehat{\theta}_t - \theta_*\right\|^2_{\frac{X_t X_t^\top}{\sigma_t^2}} \leq \left\langle \nabla\widetilde{\ell}_t(\widehat{\theta}_t), \widehat{\theta}_t - \theta_* \right\rangle$, then we have

$$\frac{1}{2}\left\|\widehat{\theta}_t - \theta_*\right\|^2_{\frac{X_t X_t^\top}{\sigma_t^2}} \leq \left\langle \nabla\ell_t(\widehat{\theta}_t), \widehat{\theta}_{t+1} - \theta_* \right\rangle + \left\langle \nabla\ell_t(\widehat{\theta}_t), \widehat{\theta}_t - \widehat{\theta}_{t+1} \right\rangle + \left\langle \nabla\widetilde{\ell}_t(\widehat{\theta}_t) - \nabla\ell_t(\widehat{\theta}_t), \widehat{\theta}_t - \theta_* \right\rangle, \quad (21)$$

where the first term can be bounded by the Bregman proximal inequality (Lemma 9), we have

$$\left\langle \nabla\ell_t(\widehat{\theta}_t), \widehat{\theta}_{t+1} - \theta_* \right\rangle \leq \mathcal{D}_{\psi_t}(\theta_*, \widehat{\theta}_t) - \mathcal{D}_{\psi_t}(\theta_*, \widehat{\theta}_{t+1}) - \mathcal{D}_{\psi_t}(\widehat{\theta}_{t+1}, \widehat{\theta}_t)$$

$$= \frac{1}{2}\left\|\widehat{\theta}_t - \theta_*\right\|^2_{V_t} - \frac{1}{2}\left\|\widehat{\theta}_{t+1} - \theta_*\right\|^2_{V_t} - \frac{1}{2}\left\|\widehat{\theta}_{t+1} - \theta_t\right\|^2_{V_t},$$

substituting the above inequality into (21), we have

$$\frac{1}{2}\left\|\widehat{\theta}_t - \theta_*\right\|^2_{\frac{X_t X_t^\top}{\sigma_t^2}} \leq \frac{1}{2}\left\|\widehat{\theta}_t - \theta_*\right\|^2_{V_t} - \frac{1}{2}\left\|\widehat{\theta}_{t+1} - \theta_*\right\|^2_{V_t} - \frac{1}{2}\left\|\widehat{\theta}_{t+1} - \theta_t\right\|^2_{V_t}$$

$$+ \left\langle \nabla\ell_t(\widehat{\theta}_t), \widehat{\theta}_t - \widehat{\theta}_{t+1} \right\rangle + \left\langle \nabla\widetilde{\ell}_t(\widehat{\theta}_t) - \nabla\ell_t(\widehat{\theta}_t), \widehat{\theta}_t - \theta_* \right\rangle. \quad (22)$$

Rearranging the above inequality and using AM-GM inequality, we have

$$\left\|\widehat{\theta}_{t+1} - \theta_*\right\|^2_{V_t} \leq \left\|\widehat{\theta}_t - \theta_*\right\|^2_{V_t} + \left\|\nabla\ell_t(\widehat{\theta}_t)\right\|^2_{V_t^{-1}} + 2\left\langle \nabla\widetilde{\ell}_t(\widehat{\theta}_t) - \nabla\ell_t(\widehat{\theta}_t), \widehat{\theta}_t - \theta_* \right\rangle - \left\|\widehat{\theta}_t - \theta_*\right\|^2_{\frac{X_t X_t^\top}{\sigma_t^2}}$$

$$\leq \left\|\widehat{\theta}_t - \theta_*\right\|^2_{V_{t-1}} + \left\|\nabla\ell_t(\widehat{\theta}_t)\right\|^2_{V_t^{-1}} + 2\left\langle \nabla\widetilde{\ell}_t(\widehat{\theta}_t) - \nabla\ell_t(\widehat{\theta}_t), \widehat{\theta}_t - \theta_* \right\rangle + \left(\frac{1}{\alpha} - 1\right)\left\|\widehat{\theta}_t - \theta_*\right\|^2_{\frac{X_t X_t^\top}{\sigma_t^2}},$$

where the last equality comes from that $V_t = V_{t-1} + \frac{1}{\alpha}\frac{X_t X_t^\top}{\sigma_t^2}$. Taking the summation of the above inequality over $t$ rounds and we have

$$\left\|\widehat{\theta}_{t+1} - \theta_*\right\|^2_{V_t}$$

$$\leq \left\|\widehat{\theta}_1 - \theta_*\right\|^2_{V_0} + \sum_{s=1}^{t}\left\|\nabla\ell_s(\widehat{\theta}_s)\right\|^2_{V_s^{-1}} + 2\sum_{s=1}^{t}\left\langle \nabla\widetilde{\ell}_s(\widehat{\theta}_s) - \nabla\ell_s(\widehat{\theta}_s), \widehat{\theta}_s - \theta_* \right\rangle + \left(\frac{1}{\alpha} - 1\right)\sum_{s=1}^{t}\left\|\widehat{\theta}_s - \theta_*\right\|^2_{\frac{X_s X_s^\top}{\sigma_s^2}}$$

$$\leq 4\lambda S^2 + \sum_{s=1}^{t}\left\|\nabla\ell_s(\widehat{\theta}_s)\right\|^2_{V_s^{-1}} + 2\sum_{s=1}^{t}\left\langle \nabla\widetilde{\ell}_s(\widehat{\theta}_s) - \nabla\ell_s(\widehat{\theta}_s), \widehat{\theta}_s - \theta_* \right\rangle + \left(\frac{1}{\alpha} - 1\right)\sum_{s=1}^{t}\left\|\widehat{\theta}_s - \theta_*\right\|^2_{\frac{X_s X_s^\top}{\sigma_s^2}},$$

where the last inequality comes from $V_0 = \lambda I_d$ and Assumption 1, thus we complete the proof. $\qquad\square$

## B.3. Proof of Lemma 4

*Proof.* We first analyze the upper bound of single term $\|\nabla\ell_t(\widehat{\theta}_t)\|^2_{V_t^{-1}}$, based on the definition of loss function (13), we have

$$\left\|\nabla\ell_t(\widehat{\theta}_t)\right\|^2_{V_t^{-1}} = \left\|\min\left\{\left|\frac{r_t - X_t^\top\widehat{\theta}_t}{\sigma_t}\right|, \tau_t\right\}\frac{X_t}{\sigma_t}\right\|^2_{V_t^{-1}}$$

$$= \left\|\min\left\{\left|\frac{\eta_t}{\sigma_t} + \frac{X_t^\top\theta_* - X_t^\top\widehat{\theta}_t}{\sigma_t}\right|, \tau_t\right\}\frac{X_t}{\sigma_t}\right\|^2_{V_t^{-1}}$$

$$\leq \left(\min\left\{\left|\frac{\eta_t}{\sigma_t}\right| + \left|\frac{X_t^\top\theta_* - X_t^\top\widehat{\theta}_t}{\sigma_t}\right|, \tau_t\right\}\right)^2\left\|\frac{X_t}{\sigma_t}\right\|^2_{V_t^{-1}}$$

$$\leq \left(\min\left\{\left|\frac{\eta_t}{\sigma_t}\right|, \tau_t\right\} + \min\left\{\left|\frac{X_t^\top\theta_* - X_t^\top\widehat{\theta}_t}{\sigma_t}\right|, \tau_t\right\}\right)^2\left\|\frac{X_t}{\sigma_t}\right\|^2_{V_t^{-1}}$$

$$\leq 2\left(\min\left\{\left|\frac{\eta_t}{\sigma_t}\right|,\tau_t\right\}\right)^2\left\|\frac{X_t}{\sigma_t}\right\|_{V_t^{-1}}^2 + 2\left(\frac{X_t^\top\theta_* - X_t^\top\widehat{\theta}_t}{\sigma_t}\right)^2\left\|\frac{X_t}{\sigma_t}\right\|_{V_t^{-1}}^2. \tag{23}$$

We define $\widetilde{V}_t \triangleq \lambda\alpha I + \sum_{s=1}^t \frac{X_s X_s^\top}{\sigma_s^2} = \alpha V_t$, then we have $w_t = \frac{1}{\sqrt{\alpha}}\left\|\frac{X_t}{\sigma_t}\right\|_{V_{t-1}^{-1}} = \left\|\frac{X_t}{\sigma_t}\right\|_{\widetilde{V}_{t-1}^{-1}}$. Based on Sherman-Morrison-Woodbury formula of $\widetilde{V}_t^{-1}$ we have

$$\left\|\frac{X_t}{\sigma_t}\right\|_{\widetilde{V}_t^{-1}}^2 = \frac{X_t^\top \widetilde{V}_t^{-1} X_t}{\sigma_t^2} = \frac{1}{\sigma_t^2}X_t^\top\left(\widetilde{V}_{t-1}^{-1} - \frac{\widetilde{V}_{t-1}^{-1}X_t X_t^\top \widetilde{V}_{t-1}^{-1}}{\sigma_t^2(1+w_t^2)}\right)X_t = w_t^2 - \frac{w_t^4}{1+w_t^2} = \frac{w_t^2}{1+w_t^2},$$

thus $\left\|\frac{X_t}{\sigma_t}\right\|_{V_t^{-1}}^2 = \alpha\frac{w_t^2}{1+w_t^2}$, then taking the summation over inequality (23) over $t$ rounds and we have

$$\sum_{s=1}^t \left\|\nabla\ell_s(\widehat{\theta}_s)\right\|_{V_s^{-1}}^2 \leq \underbrace{2\alpha\sum_{s=1}^t\left(\min\left\{\left|\frac{\eta_s}{\sigma_s}\right|,\tau_s\right\}\right)^2\frac{w_s^2}{1+w_s^2}}_{\text{TERM (A.1)}} + \underbrace{2\alpha\sum_{s=1}^t\left(\frac{X_s^\top\theta_* - X_s^\top\widehat{\theta}_s}{\sigma_s}\right)^2\frac{w_s^2}{1+w_s^2}}_{\text{TERM (A.2)}}.$$

**For TERM (A.1).** By appling Lemma C.5 of (Huang et al., 2023), restated in Lemma 6, with $\tau_t = \tau_0\frac{\sqrt{1+w_t^2}}{w_t}t^{\frac{1-\varepsilon}{2(1+\varepsilon)}}$ and $b = \max_{t\in[T]}\frac{\nu_t}{\sigma_t} \leq 1$, we have with probability at least $1-\delta, \forall t \geq 1$,

$$\sum_{s=1}^t\left(\min\left\{\left|\frac{\eta_s}{\sigma_s}\right|,\tau_s\right\}\right)^2\frac{w_s^2}{1+w_s^2} \leq \left[t^{\frac{1-\varepsilon}{2(1+\varepsilon)}}\left(\sqrt{\tau_0^{1-\varepsilon}(\sqrt{2\kappa})^{1+\varepsilon}(\log 3T)^{\frac{1-\varepsilon}{2}}} + \tau_0\sqrt{2\log\frac{2T^2}{\delta}}\right)\right]^2,$$

by choosing $\tau_0 = \frac{\sqrt{2\kappa}(\log 3T)^{\frac{1-\varepsilon}{2(1+\varepsilon)}}}{\left(\log\frac{2T^2}{\delta}\right)^{\frac{1}{1+\varepsilon}}}$, we have $\sqrt{\tau_0^{1-\varepsilon}(\sqrt{2\kappa})^{1+\varepsilon}(\log 3T)^{\frac{1-\varepsilon}{2}}} = \tau_0\sqrt{\log\frac{2T^2}{\delta}}$, which means

$$\sum_{s=1}^t\left(\min\left\{\left|\frac{\eta_s}{\sigma_s}\right|,\tau_s\right\}\right)^2\frac{w_s^2}{1+w_s^2} \leq 6\left(t^{\frac{1-\varepsilon}{2(1+\varepsilon)}}\sqrt{2\kappa}(\log 3T)^{\frac{1-\varepsilon}{2(1+\varepsilon)}}\left(\log\frac{2T^2}{\delta}\right)^{\frac{\varepsilon}{1+\varepsilon}}\right)^2. \tag{24}$$

**For TERM (A.2).** Since $\sigma_t \geq 2\sqrt{2}\|X_t\|_{V_{t-1}^{-1}}$, which means $\alpha w_t^2 = \left\|\frac{X_t}{\sigma_t}\right\|_{V_{t-1}^{-1}}^2 \leq 1/8$, then we have

$$2\alpha\sum_{s=1}^t\left(\frac{X_s^\top\theta_* - X_s^\top\widehat{\theta}_s}{\sigma_s}\right)^2\frac{w_s^2}{1+w_s^2} \leq \frac{1}{4}\sum_{s=1}^t\left(\frac{X_s^\top\theta_* - X_s^\top\widehat{\theta}_s}{\sigma_s}\right)^2 = \frac{1}{4}\sum_{s=1}^t\left\|\theta_* - \widehat{\theta}_s\right\|_{\frac{X_s X_s^\top}{\sigma^2}}^2, \tag{25}$$

Combining (24) and (25), we have with probability at least $1-\delta, \forall t \geq 1$

$$\sum_{s=1}^t\left\|\nabla\ell_s(\widehat{\theta}_s)\right\|_{V_s^{-1}}^2 \leq 6\alpha\left(t^{\frac{1-\varepsilon}{2(1+\varepsilon)}}\sqrt{2\kappa}(\log 3T)^{\frac{1-\varepsilon}{2(1+\varepsilon)}}\left(\log\frac{2T^2}{\delta}\right)^{\frac{\varepsilon}{1+\varepsilon}}\right)^2 + \frac{1}{4}\sum_{s=1}^t\left\|\theta_* - \widehat{\theta}_s\right\|_{\frac{X_s X_s^\top}{\sigma^2}}^2.$$

Thus we complete the proof. $\qquad\square$

### B.4. Proof of Lemma 5

*Proof.* We first analyze the upper bound of single term,

$$\left\langle\nabla\widetilde{\ell}_t(\widehat{\theta}_t) - \nabla\ell_t(\widehat{\theta}_t),\widehat{\theta}_t - \theta_*\right\rangle\mathbb{1}_{A_t}$$

$$= \left\langle\nabla\widetilde{\ell}_t(\widehat{\theta}_t) - \nabla\ell_t(\theta_*) + \nabla\ell_t(\theta_*) - \nabla\ell_t(\widehat{\theta}_t),\widehat{\theta}_t - \theta_*\right\rangle\mathbb{1}_{A_t}$$

$$= \underbrace{\left\langle\nabla\widetilde{\ell}_t(\widehat{\theta}_t) + \nabla\ell_t(\theta_*) - \nabla\ell_t(\widehat{\theta}_t),\widehat{\theta}_t - \theta_*\right\rangle\mathbb{1}_{A_t}}_{\text{TERM (B.1)}} + \underbrace{\left\langle-\nabla\ell_t(\theta_*),\widehat{\theta}_t - \theta_*\right\rangle\mathbb{1}_{A_t}}_{\text{TERM (B.2)}}.$$

**For TERM (B.1).** We define $\psi_t(z)$ as the gradient of Huber loss function (5), then we have

$$
\psi_t(z) = \begin{cases} z & \text{if } |z| \le \tau_t, \\ \tau_t & \text{if } z > \tau_t, \\ -\tau_t & \text{if } z < -\tau_t. \end{cases} \tag{26}
$$

Based on (12) and (14), $\nabla \ell_t(\theta) = -\psi_t(z_t(\theta))\frac{X_t}{\sigma_t}$ and $\nabla \widetilde{\ell}_t(\theta) = -\psi_t(z_t(\theta) - z_t(\theta_*))\frac{X_t}{\sigma_t}$, which means

$$
\left\langle \nabla \widetilde{\ell}_t(\widehat{\theta}_t) + \nabla \ell_t(\theta_*) - \nabla \ell_t(\widehat{\theta}_t), \widehat{\theta}_t - \theta_* \right\rangle \mathbb{1}_{A_t}
$$

$$
= \left( \psi_t(z_t(\widehat{\theta}_t)) - \psi_t(z_t(\theta_*)) - \psi_t(z_t(\widehat{\theta}_t) - z_t(\theta_*)) \right) \frac{\left\langle X_t, \widehat{\theta}_t - \theta_* \right\rangle}{\sigma_t} \mathbb{1}_{A_t},
$$

we first analyze term $\psi_t(z_t(\widehat{\theta}_t)) - \psi_t(z_t(\theta_*)) - \psi_t(z_t(\widehat{\theta}_t) - z_t(\theta_*))$. When event $A_t$ holds, similar to (18), we have

$$
\left| z_t(\widehat{\theta}_t) - z_t(\theta_*) \right| = \left| \frac{X_t^\top \widehat{\theta}_t - X_t^\top \theta_*}{\sigma_t} \right| \le \frac{\tau_t}{2},
$$

then based on (26), we have $\psi_t(z_t(\widehat{\theta}_t) - z_t(\theta_*)) = z_t(\widehat{\theta}_t) - z_t(\theta_*)$. Next, we analyze the different situation of $z_t(\theta_*)$. When $|z_t(\theta_*)| \le \frac{\tau_t}{2}$, we have

$$
\left| z_t(\widehat{\theta}_t) \right| = \left| z_t(\widehat{\theta}_t) - z_t(\theta_*) + z_t(\theta_*) \right| \le \left| z_t(\widehat{\theta}_t) - z_t(\theta_*) \right| + |z_t(\theta_*)| \le \frac{\tau_t}{2} + \frac{\tau_t}{2} = \tau_t, \tag{27}
$$

based on (26) we have

$$
\psi_t(z_t(\widehat{\theta}_t)) - \psi_t(z_t(\theta_*)) - \psi_t(z_t(\widehat{\theta}_t) - z_t(\theta_*)) = z_t(\widehat{\theta}_t) - z_t(\theta_*) - z_t(\widehat{\theta}_t) + z_t(\theta_*) = 0.
$$

For another situation such that $|z_t(\theta_*)| > \frac{\tau_t}{2}$, based on (26) we have

$$
\psi_t(z_t(\widehat{\theta}_t)) - \psi_t(z_t(\theta_*)) - \psi_t(z_t(\widehat{\theta}_t) - z_t(\theta_*))
$$
$$
\le \left| \psi_t(z_t(\widehat{\theta}_t)) \right| + |\psi_t(z_t(\theta_*))| + \left| \psi_t(z_t(\widehat{\theta}_t) - z_t(\theta_*)) \right| \le 3\tau_t. \tag{28}
$$

Combine this two situations (27) and (28), we have for TERM (B.1),

$$
\left\langle \nabla \widetilde{\ell}_t(\widehat{\theta}_t) + \nabla \ell_t(\theta_*) - \nabla \ell_t(\widehat{\theta}_t), \widehat{\theta}_t - \theta_* \right\rangle \mathbb{1}_{A_t}
$$

$$
= \left( \psi_t(z_t(\widehat{\theta}_t)) - \psi_t(z_t(\theta_*)) - \psi_t(z_t(\widehat{\theta}_t) - z_t(\theta_*)) \right) \frac{\left\langle X_t, \widehat{\theta}_t - \theta_* \right\rangle}{\sigma_t} \mathbb{1}_{A_t}
$$

$$
\le \left| \psi_t(z_t(\widehat{\theta}_t)) - \psi_t(z_t(\theta_*)) - \psi_t(z_t(\widehat{\theta}_t) - z_t(\theta_*)) \right| \left\| \frac{X_t}{\sigma_t} \right\|_{V_{t-1}^{-1}} \left\| \widehat{\theta}_t - \theta_* \right\|_{V_{t-1}} \mathbb{1}_{A_t}
$$

$$
\le \mathbb{1}\left\{ |z_t(\theta_*)| \le \frac{\tau_t}{2} \right\} 0 + \mathbb{1}\left\{ |z_t(\theta_*)| > \frac{\tau_t}{2} \right\} 3\tau_t \left\| \frac{X_t}{\sigma_t} \right\|_{V_{t-1}^{-1}} \left\| \widehat{\theta}_t - \theta_* \right\|_{V_{t-1}} \mathbb{1}_{A_t}
$$

$$
\le \mathbb{1}\left\{ |z_t(\theta_*)| > \frac{\tau_t}{2} \right\} 3\tau_0 \frac{\sqrt{1+w_t^2}}{w_t} t^{\frac{1-\varepsilon}{2(1+\varepsilon)}} \sqrt{\alpha} w_t \beta_{t-1}
$$

$$
= \mathbb{1}\left\{ |z_t(\theta_*)| > \frac{\tau_t}{2} \right\} 3\tau_0 \sqrt{\alpha + \alpha w_t^2} t^{\frac{1-\varepsilon}{2(1+\varepsilon)}} \beta_{t-1}
$$

$$
\le \mathbb{1}\left\{ |z_t(\theta_*)| > \frac{\tau_t}{2} \right\} 3\sqrt{\alpha + \frac{1}{8}} \tau_0 t^{\frac{1-\varepsilon}{2(1+\varepsilon)}} \beta_{t-1},
$$

where the third inequality comes from the definition of event $A_t$ and the last third inequality comes from $\sigma_t \geq 2\sqrt{2} \left\| X_t \right\|_{V_{t-1}^{-1}}$, such that $\alpha w_t^2 = \left\| \frac{X_t}{\sigma_t} \right\|_{V_{t-1}^{-1}}^2 \leq 1/8$. Then sum up for $t$ round, and we have

$$\sum_{s=1}^t \left\langle \nabla \widetilde{\ell}_s(\widehat{\theta}_s) + \nabla \ell_s(\theta_*) - \nabla \ell_t(\widehat{\theta}_s), \widehat{\theta}_s - \theta_* \right\rangle \mathbb{1}_{A_s} \leq 3\sqrt{\alpha + \frac{1}{8}} \tau_0 \sum_{s=1}^t s^{\frac{1-\varepsilon}{2(1+\varepsilon)}} \beta_{s-1} \mathbb{1}\left\{ |z_s(\theta_*)| > \frac{\tau_s}{2} \right\}$$

$$\leq 3\sqrt{\alpha + \frac{1}{8}} \tau_0 t^{\frac{1-\varepsilon}{2(1+\varepsilon)}} \left( \max_{s \in [t+1]} \beta_s \right) \sum_{s=1}^t \mathbb{1}\left\{ |z_s(\theta_*)| > \frac{\tau_s}{2} \right\}.$$

where the last inequality comes from that $\max_{s \in [t]} \beta_{s-1} \leq \max_{s \in [t+1]} \beta_{s-1}$ Same as Eq. (C.12) of Huang et al. (2023), by setting $\tau_0$ as (17), with probability at least $1 - \delta$, for all $t \geq 1$, we have

$$\sum_{s=1}^t \mathbb{1}\left\{ |z_s(\theta_*)| > \frac{\tau_s}{2} \right\} \leq \frac{23}{3} \log \frac{2T^2}{\delta},$$

then we have for TERM (B.1), with probability at least $1 - \delta$, for all $t \geq 1$,

$$\sum_{s=1}^t \left\langle \nabla \widetilde{\ell}_s(\widehat{\theta}_s) + \nabla \ell_s(\theta_*) - \nabla \ell_t(\widehat{\theta}_s), \widehat{\theta}_s - \theta_* \right\rangle \mathbb{1}_{A_t} \leq 23\sqrt{\alpha + \frac{1}{8}} \log \frac{2T^2}{\delta} \tau_0 t^{\frac{1-\varepsilon}{2(1+\varepsilon)}} \left( \max_{s \in [t+1]} \beta_{s-1} \right). \qquad (29)$$

**For TERM (B.2).** We first have

$$\sum_{s=1}^t \left\langle -\nabla \ell_s(\theta_*), \widehat{\theta}_s - \theta_* \right\rangle \mathbb{1}_{A_s} = \sum_{s=1}^t \psi_s(z_s(\theta_*)) \left\langle \frac{X_s}{\sigma_s}, \widehat{\theta}_s - \theta_* \right\rangle \mathbb{1}_{A_s}$$

$$\leq \left| \sum_{s=1}^t \psi_s(z_s(\theta_*)) \left\langle \frac{X_s}{\sigma_s}, \widehat{\theta}_s - \theta_* \right\rangle \right|, \qquad (30)$$

where the last inequality comes from that $\mathbb{1}_{A_s} \leq 1$. Then, Lemma C.2 of Huang et al. (2023) (Self-normalized concentration) shows that by setting $\tau_0$ as (17) and $b = \max_{t \in [T]} \frac{\nu_t}{\sigma_t} \leq 1$, we have with probability at least $1 - 2\delta$, $\forall t \geq 1$,

$$\left\| \sum_{s=1}^t \psi_s(z_s(\theta_*)) \frac{X_s}{\sigma_s} \right\|_{\widetilde{V}_t^{-1}} \leq 8 t^{\frac{1-\varepsilon}{2(1+\varepsilon)}} \sqrt{2\kappa} (\log 3T)^{\frac{1-\varepsilon}{2(1+\varepsilon)}} \left( \log\left( \frac{2T^2}{\delta} \right) \right)^{\frac{\varepsilon}{1+\varepsilon}}. \qquad (31)$$

where $\widetilde{V}_t = \lambda \alpha I + \sum_{s=1}^t \frac{X_s X_s^\top}{\sigma_s}$ and $\kappa = d \log\left( 1 + \frac{L^2 T}{\sigma_{\min}^2 \lambda \alpha d} \right)$. Now we need to convert it into a 1-dimensional version, if $Z_s$ is scalar, we have:

$$\left\| \sum_{s=1}^t \psi_s(z_s(\theta_*)) \frac{Z_s}{\sigma_s} \right\|_{\widetilde{V}_t^{-1}}^2 = \frac{\left( \sum_{s=1}^t \psi_s(z_s(\theta_*)) \frac{Z_s^2}{\sigma_s^2} \right)}{\lambda \alpha + \sum_{s=1}^t \frac{Z_s^2}{\sigma_s^2}},$$

Based on inequality (31), we have

$$\frac{\left( \sum_{s=1}^t \psi_s(z_s(\theta_*)) \frac{Z_s}{\sigma_s} \right)^2}{\lambda \alpha + \sum_{s=1}^t \frac{Z_s^2}{\sigma_s^2}} \leq \left( 8 t^{\frac{1-\varepsilon}{2(1+\varepsilon)}} \sqrt{2\kappa} (\log 3T)^{\frac{1-\varepsilon}{2(1+\varepsilon)}} \left( \log\left( \frac{2T^2}{\delta} \right) \right)^{\frac{\varepsilon}{1+\varepsilon}} \right)^2$$

$$\left| \sum_{s=1}^t \psi_s(z_s(\theta_*)) \frac{Z_s}{\sigma_s} \right| \leq \sqrt{\left( \lambda \alpha + \sum_{s=1}^t \frac{Z_s^2}{\sigma_s^2} \right) \left( 8 t^{\frac{1-\varepsilon}{2(1+\varepsilon)}} \sqrt{2\kappa} (\log 3T)^{\frac{1-\varepsilon}{2(1+\varepsilon)}} \left( \log\left( \frac{2T^2}{\delta} \right) \right)^{\frac{\varepsilon}{1+\varepsilon}} \right)^2}.$$

Then based on AM-GM, we have

$$\left| \sum_{s=1}^t \psi_s(z_s(\theta_*)) \frac{Z_s}{\sigma_s} \right| \leq \frac{1}{4} \left( \lambda \alpha + \sum_{s=1}^t \frac{Z_s^2}{\sigma_s^2} \right) + \left( 8 t^{\frac{1-\varepsilon}{2(1+\varepsilon)}} \sqrt{2\kappa} (\log 3T)^{\frac{1-\varepsilon}{2(1+\varepsilon)}} \left( \log\left( \frac{2T^2}{\delta} \right) \right)^{\frac{\varepsilon}{1+\varepsilon}} \right)^2,$$

where we use $\sqrt{ab} \leq \frac{1}{4}a + b$. Let $Z_s = \left\langle X_s, \widehat{\theta}_s - \theta_* \right\rangle$, then put it back to (30), for TERM (2), we have with probability at least $1 - 2\delta, \forall t \geq 1$,

$$\sum_{s=1}^{t} \left\langle -\nabla \ell_s(\theta_*), \widehat{\theta}_s - \theta_* \right\rangle \leq \frac{1}{4} \left( \lambda \alpha + \sum_{s=1}^{t} \left\langle \frac{X_s}{\sigma_s}, \widehat{\theta}_s - \theta_* \right\rangle^2 \right)$$
$$+ \left( 8t^{\frac{1-\varepsilon}{2(1+\varepsilon)}} \sqrt{2\kappa}(\log 3T)^{\frac{1-\varepsilon}{2(1+\varepsilon)}} \left( \log \frac{2T^2}{\delta} \right)^{\frac{\varepsilon}{1+\varepsilon}} \right)^2 . \tag{32}$$

Combine (29) and (32) together, with union bound we have with probability at least $1 - 3\delta, \forall t \geq 1$

$$\sum_{s=1}^{t} \left\langle \nabla \widetilde{\ell}_s(\widehat{\theta}_s) - \nabla \ell_s(\widehat{\theta}_s), \widehat{\theta}_s - \theta_* \right\rangle \mathbb{1}_{A_s} \leq 23\sqrt{\alpha + \frac{1}{8} \log \frac{2T^2}{\delta}} \tau_0 t^{\frac{1-\varepsilon}{2(1+\varepsilon)}} \left( \max_{s \in [t+1]} \beta_{s-1} \right)$$
$$+ \frac{1}{4} \left( \lambda \alpha + \sum_{s=1}^{t} \left\langle \frac{X_s}{\sigma_s}, \widehat{\theta}_s - \theta_* \right\rangle^2 \right) + \left( 8t^{\frac{1-\varepsilon}{2(1+\varepsilon)}} \sqrt{2\kappa}(\log 3T)^{\frac{1-\varepsilon}{2(1+\varepsilon)}} \left( \log \frac{2T^2}{\delta} \right)^{\frac{\varepsilon}{1+\varepsilon}} \right)^2 .$$

$\square$

### B.5. Proof of Lemma 1

*Proof.* Combining the results of Lemma 4 and Lemma 5, with applying the union bound, we obtain that, with probablity at least $1 - 4\delta$, the following holds for all $t \geq 1$,

$$4\lambda S^2 + \sum_{s=1}^{t} \left\| \nabla \ell_s(\widehat{\theta}_s) \right\|_{V_s^{-1}}^2 + 2 \sum_{s=1}^{t} \left\langle \nabla \widetilde{\ell}_s(\widehat{\theta}_s) - \nabla \ell_s(\widehat{\theta}_s), \widehat{\theta}_s - \theta_* \right\rangle \mathbb{1}_{A_s} + \left( \frac{1}{\alpha} - 1 \right) \sum_{s=1}^{t} \left\| \widehat{\theta}_s - \theta_* \right\|_{\frac{X_s X_s^\top}{\sigma_s^2}}^2$$

$$\leq 4\lambda S^2 + 6\alpha \left( t^{\frac{1-\varepsilon}{2(1+\varepsilon)}} \sqrt{2\kappa}(\log 3T)^{\frac{1-\varepsilon}{2(1+\varepsilon)}} \left( \log \frac{2T^2}{\delta} \right)^{\frac{\varepsilon}{1+\varepsilon}} \right)^2 + \frac{1}{4} \sum_{s=1}^{t} \left\| \theta_* - \widehat{\theta}_s \right\|_{\frac{X_s X_s^\top}{\sigma^2}}^2$$

$$+ 46\sqrt{\alpha + \frac{1}{8} \log \frac{2T^2}{\delta}} \tau_0 t^{\frac{1-\varepsilon}{2(1+\varepsilon)}} \left( \max_{s \in [t+1]} \beta_{s-1} \right) + \frac{1}{2} \left( \lambda \alpha + \sum_{s=1}^{t} \left\langle \frac{X_s}{\sigma_s}, \widehat{\theta}_s - \theta_* \right\rangle^2 \right)$$

$$+ 2 \left( 8t^{\frac{1-\varepsilon}{2(1+\varepsilon)}} \sqrt{2\kappa}(\log 3T)^{\frac{1-\varepsilon}{2(1+\varepsilon)}} \left( \log \frac{2T^2}{\delta} \right)^{\frac{\varepsilon}{1+\varepsilon}} \right)^2 + \left( \frac{1}{\alpha} - 1 \right) \sum_{s=1}^{t} \left\| \widehat{\theta}_s - \theta_* \right\|_{\frac{X_s X_s^\top}{\sigma_s^2}}^2$$

$$\leq 94 \log \frac{2T^2}{\delta} \tau_0 t^{\frac{1-\varepsilon}{2(1+\varepsilon)}} \left( \max_{s \in [t+1]} \beta_{s-1} \right) + \lambda(2 + 4S^2) + 152 \left( \log \frac{2T^2}{\delta} \tau_0 t^{\frac{1-\varepsilon}{2(1+\varepsilon)}} \right)^2 ,$$

where the least inequality comes from that $\alpha = 4$, and

$$\log \frac{2T^2}{\delta} \tau_0 t^{\frac{1-\varepsilon}{2(1+\varepsilon)}} = t^{\frac{1-\varepsilon}{2(1+\varepsilon)}} \sqrt{2\kappa}(\log 3T)^{\frac{1-\varepsilon}{2(1+\varepsilon)}} \left( \log \frac{2T^2}{\delta} \right)^{\frac{\varepsilon}{1+\varepsilon}} .$$

Then we choose $\forall t \geq 1$,

$$\beta_t = 107 \log \frac{2T^2}{\delta} \tau_0 t^{\frac{1-\varepsilon}{2(1+\varepsilon)}} + \sqrt{\lambda (2 + 4S^2)},$$

and we can varify that with probablity at least $1 - 4\delta$, the following holds for all $t \geq 1$,

$$\beta_t^2 \geq 4\lambda S^2 + \sum_{s=1}^{t} \left\| \nabla \ell_s(\widehat{\theta}_s) \right\|_{V_s^{-1}}^2 + 2 \sum_{s=1}^{t} \left\langle \nabla \widetilde{\ell}_s(\widehat{\theta}_s) - \nabla \ell_s(\widehat{\theta}_s), \widehat{\theta}_s - \theta_* \right\rangle \mathbb{1}_{A_s}$$
$$+ \left( \frac{1}{\alpha} - 1 \right) \sum_{s=1}^{t} \left\| \widehat{\theta}_s - \theta_* \right\|_{\frac{X_s X_s^\top}{\sigma_s^2}}^2 . \tag{33}$$

Let $B$ denote the event that the conditions in (33) hold $\forall t \geq 1$, then $\mathbb{P}(B) \geq 1 - 4\delta$. We now introduce a new event $C$ that is define as

$$C \triangleq \left\{ \forall t \geq 1, \left\| \widehat{\theta}_t - \theta_* \right\|_{V_{t-1}} \leq \beta_{t-1} \right\} = \bigcap_{t=0}^{\infty} A_t.$$

In the following we will show that if $B$ is true, $C$ must be true, which means $\mathbb{P}(C) \geq \mathbb{P}(B) \geq 1 - 4\delta$ by mathematical induction. When $t = 1$, $A_1$ is true by definition such that $\left\| \widehat{\theta}_1 - \theta_* \right\|_{V_0} \leq \sqrt{4\lambda S^2} \leq \sqrt{\lambda(2 + 4S^2)} = \beta_0$. Suppose that at iteration $t$, for all $s \in [t]$, $A_s$ is true, then we are going to show that $A_{t+1}$ is also true.

$$\left\| \widehat{\theta}_{t+1} - \theta_* \right\|_{V_t}^2$$

$$\leq 4\lambda S^2 + \sum_{s=1}^{t} \left\| \nabla \ell_s(\widehat{\theta}_s) \right\|_{V_s^{-1}}^2 + 2 \sum_{s=1}^{t} \left\langle \nabla \widetilde{\ell}_s(\widehat{\theta}_s) - \nabla \ell_s(\widehat{\theta}_s), \widehat{\theta}_s - \theta_* \right\rangle + \left( \frac{1}{\alpha} - 1 \right) \sum_{s=1}^{t} \left\| \widehat{\theta}_s - \theta_* \right\|_{\frac{X_s X_s^\top}{\sigma_s^2}}^2$$

$$= 4\lambda S^2 + \sum_{s=1}^{t} \left\| \nabla \ell_s(\widehat{\theta}_s) \right\|_{V_s^{-1}}^2 + 2 \sum_{s=1}^{t} \left\langle \nabla \widetilde{\ell}_s(\widehat{\theta}_s) - \nabla \ell_s(\widehat{\theta}_s), \widehat{\theta}_s - \theta_* \right\rangle \mathbb{1}_{A_s} + \left( \frac{1}{\alpha} - 1 \right) \sum_{s=1}^{t} \left\| \widehat{\theta}_s - \theta_* \right\|_{\frac{X_s X_s^\top}{\sigma_s^2}}^2$$

$$\leq \beta_t^2,$$

where the first inequality comes from Lemma 3, the second equality comes from that $\forall s \in [t]$, $A_s$ holds, and the last inequality comes from condition (33). As a result, we can conclude that all $\{A_t\}_{t \geq 1}$ is true and thus we have $\mathbb{P}(C) \geq 1 - 4\delta$. And further we can find that

$$\sqrt{\frac{2 \|X_t\|_{V_{t-1}^{-1}}^2 \beta_{t-1}}{\sqrt{\alpha} \tau_0 t^{\frac{1-\varepsilon}{2(1+\varepsilon)}}}} = \sqrt{\frac{107 \log \frac{2T^2}{\delta} \tau_0 t^{\frac{1-\varepsilon}{2(1+\varepsilon)}} + \sqrt{\lambda(2 + 4S^2)}}{\tau_0 t^{\frac{1-\varepsilon}{2(1+\varepsilon)}}}} \|X_t\|_{V_{t-1}^{-1}} \geq \sqrt{107} \|X_t\|_{V_{t-1}^{-1}} \geq 2\sqrt{2} \|X_t\|_{V_{t-1}^{-1}},$$

thus by setting

$$\sigma_t = \max \left\{ \nu_t, \sigma_{\min}, \sqrt{\frac{\|X_t\|_{V_{t-1}^{-1}}^2 \beta_{t-1}}{\tau_0 t^{\frac{1-\varepsilon}{2(1+\varepsilon)}}}} \right\}, \quad \tau_t = \tau_0 \frac{\sqrt{1 + w_t^2}}{w_t} t^{\frac{1-\varepsilon}{2(1+\varepsilon)}}, \quad \tau_0 = \frac{\sqrt{2\kappa}(\log 3T)^{\frac{1-\varepsilon}{2(1+\varepsilon)}}}{\left( \log \frac{2T^2}{\delta} \right)^{\frac{1}{1+\varepsilon}}},$$

we obtain for any $\delta \in (0, 1)$, with probablity at least $1 - 4\delta$, the following holds for all $t \geq 1$,

$$\left\| \widehat{\theta}_{t+1} - \theta_* \right\|_{V_t} \leq 107 \log \frac{2T^2}{\delta} \tau_0 t^{\frac{1-\varepsilon}{2(1+\varepsilon)}} + \sqrt{\lambda(2 + 4S^2)}.$$

$\square$

## C. Regret Analysis

### C.1. Proof of Theorem 1

*Proof.* Let $X_t^* = \arg\max_{\mathbf{x} \in \mathcal{X}_t} \mathbf{x}^\top \theta_*$. Due to Lemma 1 and the fact that $X_t^*, X_t \in \mathcal{X}_t$, each of the following holds with probability at least $1 - 4\delta$

$$\forall t \in [T], \quad X_t^{*\top} \theta_* \leq X_t^{*\top} \widehat{\theta}_t + \beta_{t-1} \|X_t^*\|_{V_{t-1}^{-1}}$$

$$\forall t \in [T], \quad X_t^\top \theta_* \geq X_t^\top \widehat{\theta}_t - \beta_{t-1} \|X_t\|_{V_{t-1}^{-1}}.$$

By the union bound, the following holds with probability at least $1 - 8\delta$,

$$\forall t \in [T], \quad X_t^{*\top} \theta_* - X_t^\top \theta_* \leq X_t^{*\top} \widehat{\theta}_t - X_t^\top \widehat{\theta}_t + \beta_{t-1} \left( \|X_t^*\|_{V_{t-1}^{-1}} + \|X_t\|_{V_{t-1}^{-1}} \right) \leq 2\beta_{t-1} \|X_t\|_{V_{t-1}^{-1}},$$

where the last inequality comes from the arm selection criteria (11) such that

$$X_t^{*\top}\widehat{\theta}_t + \beta_{t-1}\|X_t^*\|_{V_{t-1}^{-1}} \le X_t^\top\widehat{\theta}_t + \beta_{t-1}\|X_t\|_{V_{t-1}^{-1}}.$$

Hence the following regret bound holds with probability at least $1 - 8\delta$,

$$\mathrm{REG}_T = \sum_{t=1}^T X_t^{*\top}\theta_* - \sum_{t=1}^T X_t^\top\theta_* \le 2\beta_T \sum_{t=1}^T \|X_t\|_{V_{t-1}^{-1}} = 4\beta_T \sum_{t=1}^T \sigma_t w_t,$$

where $\beta_T = 107T^{\frac{1-\varepsilon}{2(1+\varepsilon)}}\tau_0\log\frac{2T^2}{\delta} + \sqrt{\lambda(2+4S^2)}$.

Next we bound the sum of bonus $\sum_{t=1}^T \sigma_t w_t$ separately by the value of $\sigma_t$. Recall the definition of $\sigma_t$ in Algorithm 2, we decompose $[T]$ as the union of three disjoint sets $\mathcal{J}_1, \mathcal{J}_2$,

$$\mathcal{J}_1 = \{t \in [T] \mid \sigma_t \in \{\nu_t, \sigma_{\min}\}\}, \quad \mathcal{J}_2 = \left\{t \in [T] \,\middle|\, \sigma_t = \sqrt{\frac{\|X_t\|_{V_{t-1}^{-1}}^2 \beta_{t-1}}{\tau_0 t^{\frac{1-\varepsilon}{2(1+\varepsilon)}}}}\right\}. \tag{34}$$

For the summation over $\mathcal{J}_1$, since $\sigma_{\min} = \frac{1}{\sqrt{T}}$,

$$\sum_{t\in\mathcal{J}_1}\sigma_t w_t = \sum_{t\in\mathcal{J}_1}\max\{\nu_t, \sigma_{\min}\}w_t \le \sqrt{\sum_{t=1}^T(\nu_t^2 + \sigma_{\min}^2)}\sqrt{\sum_{t=1}^T w_t^2} \le \sqrt{2\kappa}\sqrt{\sum_{t=1}^T \nu_t^2 + 1},$$

where the last inequality comes from Lemma 7.

For the summation over $\mathcal{J}_2$, we first have

$$w_t^{-2} = \frac{\beta_{t-1}}{\tau_0 t^{\frac{1-\varepsilon}{2(1+\varepsilon)}}} = \frac{107t^{\frac{1-\varepsilon}{2(1+\varepsilon)}}\tau_0\log\frac{2T^2}{\delta} + \sqrt{\lambda(2+4S^2)}}{\tau_0 t^{\frac{1-\varepsilon}{2(1+\varepsilon)}}} \le \frac{\sqrt{\lambda(2+4S^2)}}{\tau_0} + 107\log\frac{2T^2}{\delta},$$

we denote $c = \frac{\sqrt{\lambda(2+4S^2)}}{\tau_0} + 107\log\frac{2T^2}{\delta}$ and have

$$\sum_{t\in\mathcal{J}_2}\sigma_t w_t = \sum_{t\in\mathcal{J}_2}\frac{1}{w_t^2}\|X_t\|_{\widetilde{V}_{t-1}^{-1}}w_t^2 \le \frac{cL}{2\sqrt{\lambda}}\sum_{t\in\mathcal{J}_2}w_t^2 \le \frac{cL\kappa}{\sqrt{\lambda}}. \tag{35}$$

where the last inequality comes from Lemma 7. Combining these two cases together, by choosing $\delta = \frac{1}{8T}$, $\lambda = d$, we have

$$\mathrm{REG}_T \le 4\beta_T\left(\sqrt{2\kappa}\sqrt{\sum_{t=1}^T \nu_t^2 + 1} + \frac{cL\kappa}{\sqrt{\lambda}}\right) \le \widetilde{\mathcal{O}}\left(\sqrt{d}T^{\frac{1-\varepsilon}{2(1+\varepsilon)}}\left(\sqrt{d\sum_{t=1}^T \nu_t^2} + \sqrt{d}\right)\right)$$

$$= \widetilde{\mathcal{O}}\left(dT^{\frac{1-\varepsilon}{2(1+\varepsilon)}}\sqrt{\sum_{t=1}^T \nu_t^2} + dT^{\frac{1-\varepsilon}{2(1+\varepsilon)}}\right).$$

Thus we complete the proof. $\qquad\square$

## C.2. Proof of Corollary 1

*Proof.* Proof of Corollary 1 is similar to Theorem 1, only need to change the condition $\mathcal{J}_1$ in (34) to $\mathcal{J}_1 = \{t \in [T] \mid \sigma_t \in \{\nu, \sigma_{\min}\}\}$ and for the summation over $\mathcal{J}_1$, since $\sigma_{\min} = \frac{1}{\sqrt{T}}$,

$$\sum_{t\in\mathcal{J}_1}\sigma_t w_t = \sum_{t\in\mathcal{J}_1}\max\{\nu, \sigma_{\min}\}w_t \le \sqrt{\sum_{t=1}^T(\nu^2 + \sigma_{\min}^2)}\sqrt{\sum_{t=1}^T w_t^2} \le \sqrt{2\kappa}\nu\sqrt{T+1}, \tag{36}$$

where the last inequality comes from Lemma 7. Then combine condition (36) with condition (35), by choosing $\delta = \frac{1}{8T}$, $\lambda = d$, we have

$$
\begin{aligned}
\text{REG}_T &\le 4\beta_T \sum_{t=1}^{T} \sigma_t w_t \le 4\beta_T \left( \sqrt{2\kappa}\nu\sqrt{T+1} + \frac{cL\kappa}{\sqrt{\lambda}} \right) \\
&\le \tilde{\mathcal{O}} \left( \sqrt{d}T^{\frac{1-\varepsilon}{2(1+\varepsilon)}} \left( \sqrt{d}\nu\sqrt{T} + \sqrt{d} \right) \right) \\
&= \tilde{\mathcal{O}} \left( d\nu T^{\frac{1}{1+\varepsilon}} \right).
\end{aligned}
$$

Thus we complete the proof. $\qquad\square$

## D. Technical Lemmas

This section contains some technical lemmas that are used in the proofs.

### D.1. Concentrations

**Theorem 2** (Self-normalized concentration for scalar (Abbasi-Yadkori et al., 2012, Lemma 7)). *Let $\{F_t\}_{t=0}^{\infty}$ be a filtration. Let $\{\eta_t\}_{t=0}^{\infty}$ be a real-valued stochastic process such that $\eta_t$ is $F_t$-measurable and $R$-sub-Gaussian. Let $\{Z_t\}_{t=1}^{\infty}$ be a sequence of real-valued variables such that $Z_t$ is $F_{t-1}$-measurable. Assume that $V > 0$ be deterministic. For any $\delta > 0$, with probability at least $1 - \delta$:*

$$
\forall t \ge 0, \quad \left| \sum_{s=1}^{t} \eta_s Z_s \right| \le R \sqrt{ 2 \left( V + \sum_{s=1}^{t} Z_s^2 \right) \ln \left( \frac{\sqrt{V + \sum_{s=1}^{t} Z_s^2}}{\delta \sqrt{V}} \right) }.
$$

**Lemma 6** (Lemma C.5 of Huang et al. (2023)). *By setting $\tau_t = \tau_0 \frac{\sqrt{1+w_t^2}}{w_t} t^{\frac{1-\varepsilon}{2(1+\varepsilon)}}$, assume $\mathbb{E}\left\{ \frac{\eta_t}{\sigma_t} \,\middle|\, \mathcal{F}_{t-1} \right\} = 0$ and $\mathbb{E}\left\{ \left| \frac{\eta_t}{\sigma_t} \right|^{1+\varepsilon} \,\middle|\, \mathcal{F}_{t-1} \right\} \le b^{1+\varepsilon}$. Then with probability at least $1 - \delta$, we have $\forall t \ge 1$,*

$$
\sum_{s=1}^{t} \left( \min \left\{ \left| \frac{\eta_s}{\sigma_s} \right|, \tau_s \right\} \right)^2 \frac{w_s^2}{1 + w_s^2} \le t^{\frac{1-\varepsilon}{1+\varepsilon}} \left( \sqrt{\tau_0^{1-\varepsilon}(\sqrt{2\kappa}b)^{1+\varepsilon}(\log 3t)^{\frac{1-\varepsilon}{2}}} + \tau_0\sqrt{2\log\frac{2t^2}{\delta}} \right)^2 .
$$

### D.2. Potential Lemma

**Lemma 7** (Elliptical Potential Lemma). *Suppose $U_0 = \lambda I$, $U_t = U_{t-1} + X_t X_t^{\top}$, and $\|X_t\|_2 \le L$, denote $\forall t \ge 1$, $\left\| U_{t-1}^{-\frac{1}{2}} X_t \right\|_2^2 \le c_{\max}$, then*

$$
\sum_{t=1}^{T} \left\| U_{t-1}^{-\frac{1}{2}} X_t \right\|_2^2 \le 2 \max\{1, c_{\max}\} d \log \left( 1 + \frac{L^2 T}{\lambda d} \right). \tag{37}
$$

*Proof.* First, we have the following decomposition,

$$
U_t = U_{t-1} + X_t X_t^{\top} = U_{t-1}^{\frac{1}{2}}(I + U_{t-1}^{-\frac{1}{2}} X_t X_t^{\top} U_{t-1}^{-\frac{1}{2}}) U_{t-1}^{\frac{1}{2}}.
$$

Taking the determinant on both sides, we get

$$
\det(U_t) = \det(U_{t-1}) \det(I + U_{t-1}^{-\frac{1}{2}} X_t X_t^{\top} U_{t-1}^{-\frac{1}{2}}),
$$

which in conjunction with Lemma 8 yields

$$
\det(U_t) = \det(U_{t-1})(1 + \|U_{t-1}^{-\frac{1}{2}} X_t\|_2^2) \ge \det(U_{t-1}) \left( 1 + \frac{1}{\max\{1, c_{\max}\}} \|U_{t-1}^{-\frac{1}{2}} X_t\|_2^2 \right)
$$

$$\geq \det(U_{t-1}) \exp\left(\frac{1}{2\max\{1, c_{\max}\}} \|U_{t-1}^{-\frac{1}{2}} X_t\|_2^2\right).$$

Note that in the last inequality, we use the fact that $\|U_{t-1}^{-\frac{1}{2}} X_t\|_2^2 \leq c_{\max}$ and $1 + x \geq \exp(x/2)$ holds for any $x \in [0, 1]$. By taking advantage of the telescope structure, we have

$$\sum_{t=1}^{T} \|U_{t-1}^{-\frac{1}{2}} X_t\|_2^2 \leq 2\max\{1, c_{\max}\} \log \frac{\det(U_T)}{\det(U_0)} \leq 2\max\{1, c_{\max}\} d \log\left(1 + \frac{L^2 T}{\lambda d}\right),$$

where the last inequality follows from the fact that $\mathrm{Tr}(U_T) \leq \mathrm{Tr}(U_0) + L^2 T = \lambda d + L^2 T$, and thus $\det(U_T) \leq (\lambda + L^2 T/d)^d$. $\square$

**Lemma 8** (Lemma 5 of Zhao et al. (2020a)). *For any* $\mathbf{v} \in \mathbb{R}^d$, *we have*

$$\det(I + \mathbf{v}\mathbf{v}^\top) = 1 + \|\mathbf{v}\|_2^2.$$

### D.3. Useful Lemma for OMD

**Lemma 9** (Bregman proximal inequality (Chen & Teboulle, 1993, Lemma 3.2)). *Let* $\mathcal{X}$ *be a convex set in a Banach space. Let* $f : \mathcal{X} \mapsto \mathbb{R}$ *be a closed proper convex function on* $\mathcal{X}$. *Given a convex regularizer* $\psi : \mathcal{X} \mapsto \mathbb{R}$, *we denote its induced Bregman divergence by* $\mathcal{D}_\psi(\cdot, \cdot)$. *Then, any update of the form*

$$\mathbf{x}_k = \arg\min_{\mathbf{x} \in \mathcal{X}} \{f(\mathbf{x}) + \mathcal{D}_\psi(\mathbf{x}, \mathbf{x}_{k-1})\},$$

*satisfies the following inequality for any* $\mathbf{u} \in \mathcal{X}$,

$$f(\mathbf{x}_k) - f(\mathbf{u}) \leq \mathcal{D}_\psi(\mathbf{u}, \mathbf{x}_{k-1}) - \mathcal{D}_\psi(\mathbf{u}, \mathbf{x}_k) - \mathcal{D}_\psi(\mathbf{x}_k, \mathbf{x}_{k-1}).$$

## E. Additional Experimental Results

In this section, we provide additional experimental results to validate the effectiveness of our algorithm. Appendix E.1 shows the experiment of different noise distributions, Appendix E.2 presents the experiment of varying $\nu_t$, and Appendix E.3 provides the experiment of varying arm set.

### E.1. Experiment of Different Noise Distribution

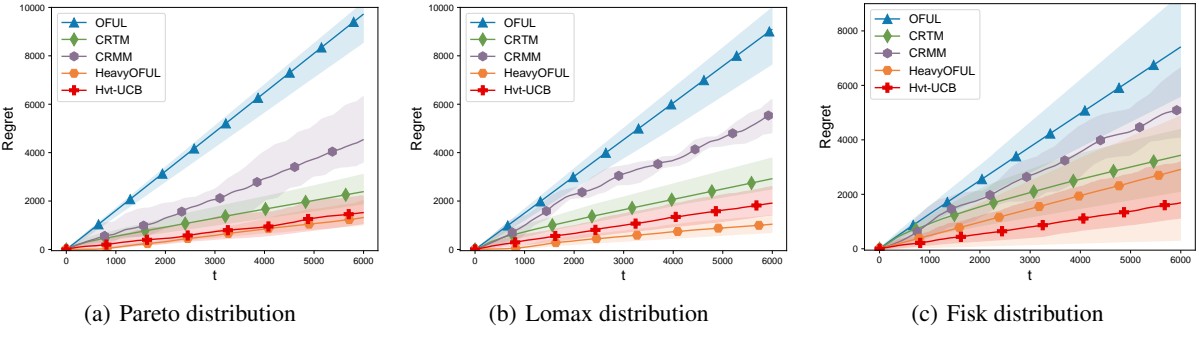

(a) Pareto distribution      (b) Lomax distribution      (c) Fisk distribution

Figure 3: Experiment of Different Noise Distribution

**Configurations.** We consider three classic heavy-tailed distributions: (a) Pareto distribution, (b) Lomax distribution, and (c) Fisk distribution. We implement all three distributions using Python's scipy.stats. For the Pareto distribution, we set $b = 1.5$; for the Lomax and Fisk distributions, we set $c = 1.5$. We set $\varepsilon = 1.49$, ensuring that the $(1 + \varepsilon)$-moment exists. In the experiments, we use a time-varying noise scale, consistent with the setup in Figure 4.

**Results.** We conduct 5 independent trials and averaged the results. Figure 3(a), 3(b) and 3(c) show cumulative regret under different noise distributions. In both settings, our algorithm performs similarly to Heavy-OFUL, demonstrating competitive performance. Note that the CRMM algorithm performs poorly across all three distributions, as it can only handle symmetric noise, while these distributions are non-symmetric. In summary, this experiment validates the effectiveness of our algorithm in handling various noise distributions.

## E.2. Experiment of Varying $\nu_t$

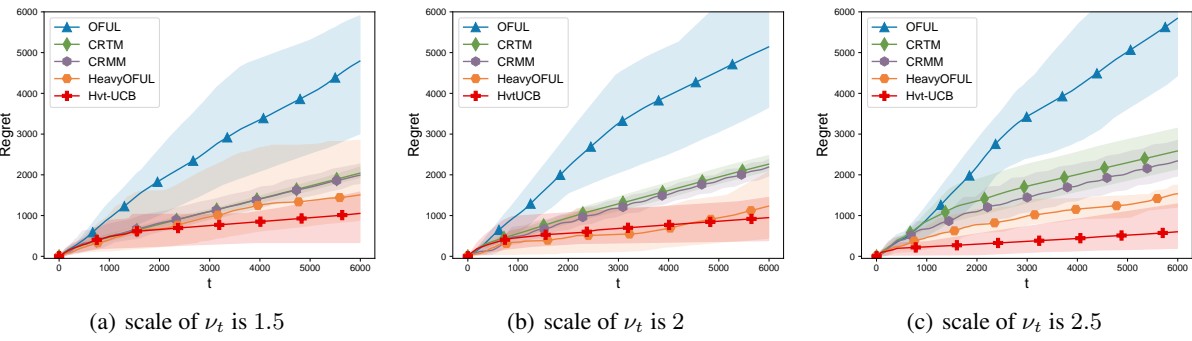

|                     |                     |                     |
|:-------------------:|:-------------------:|:-------------------:|
| (a) scale of $\nu_t$ is 1.5 | (b) scale of $\nu_t$ is 2 | (c) scale of $\nu_t$ is 2.5 |

Figure 4: Experiment of Varying $\nu_t$.

**Configurations.** In this experiment, we consider a Student's t-distribution with a time-varying noise scale. Specifically, we fix the degrees of freedom as $df = 1.7$ and set $\varepsilon = df - 0.01$. At each round, noise is first sampled from Student t$(df)$ and then scaled by a factor $\alpha$, where $\log_{10}(\alpha) \sim \text{Unif}(0, scale)$, so that the central moments of $\epsilon_t$ vary across rounds. In this setting, existing algorithms can only rely on an upper bound of the noise variance, whereas our proposed HvtLB algorithm leverages the actual per-round variance for more accurate and adaptive scheduling. This setting follows the experimental setup proposed by Huang et al. (2023).

**Results.** We test different scales of $\nu_t$, with scales of 1.5, 2 and 2.5 in Figure 4(a), 4(b) and 4(c). In all environments, HvtLB and HeavyOFUL, which have variance-aware capabilities, outperform other algorithms significantly. In contrast, OFUL performs the worst because it lacks both variance-awareness and the ability to handle heavy-tailed noise.

## E.3. Experiment of Varying Arm Set

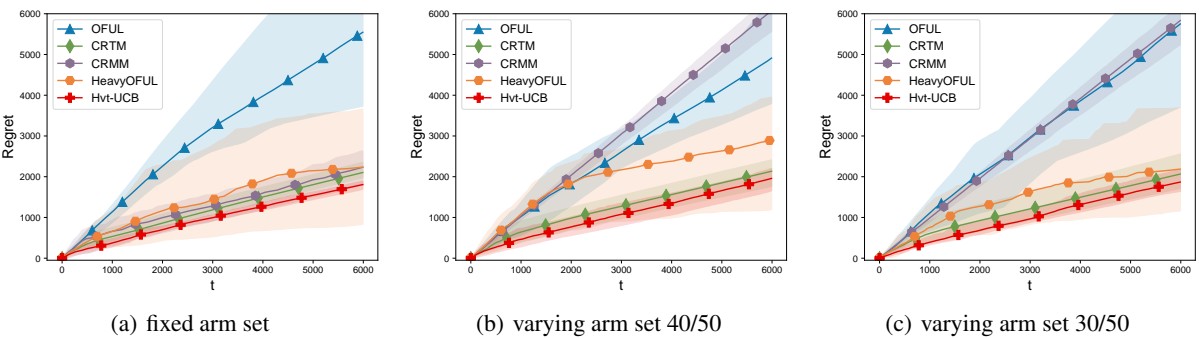

|                     |                     |                     |
|:-------------------:|:-------------------:|:-------------------:|
| (a) fixed arm set | (b) varying arm set 40/50 | (c) varying arm set 30/50 |

Figure 5: Experiment of Varying Arm Set Case.

**Configurations.** In this experiment, we create a total arm set $\mathcal{X}$ with 50 arms, which are pre-sampled before the experiment begins. To make the arm set varying, in each round, we randomly select a subset from $\mathcal{X}$ as the current arm set $\mathcal{X}_t$, from which the algorithm can choose arms. In this case, the regret is based on the best arm in each round, rather than the global optimal arm. We consider three scenarios: (i) the arm set $\mathcal{X}_t = \mathcal{X}$ with 50 arms, (ii) randomly selecting 40 arms in each round, and (iii) randomly selecting 30 arms in each round.

**Results.** We conduct 5 independent trials and averaged the results. Figure 5(a), 5(b) and 5(c) show cumulative regret under different levels of arm set changes. In all settings, our algorithm performs the best, while HeavyOFUL and CRTM also achieve competitive results. However, when the arm set varies, the MOM-based algorithm fails. This is because MOM-based algorithms assume a fixed arm set and rely on resampling. When the arm set changes, resampling the same arms is not possible, so the algorithm selects the closest available arms instead. In summary, this experiment validates the effectiveness of our algorithm in scenarios where the arm set is time-varying.

