# OpenReview forum: "Heavy-Tailed Linear Bandits: Huber Regression with One-Pass Update"
_ICML.cc/2025/Conference — ICML 2025 poster_

### Official Review · Reviewer_P7Rp · 2025-02-16

**Overall Recommendation:** 3

**Summary:**

This paper proposes a one-pass algorithm for stochastic linear bandits with heavy-tailed noise, reducing per-round computational cost from $\mathcal{O}(t \log T)$ to $\mathcal{O}(1)$ using an online mirror descent framework. Unlike existing methods that require storing and processing all past data, the proposed approach updates using only current round data while achieving a near-optimal, variance-aware regret bound. The authors also deduced the regret bound for the proposed algorithm, which is nearly optimal.

**Claims And Evidence:**

I read the technical sketch in the main paper but didn't look closely at the details in the appendix. I think the claims and theorems make sense to me.

**Essential References Not Discussed:**

I think most of the notable references have been sufficiently discussed in this work. One minor suggestion:

Since I am not familiar with Huber regression and its applications, I need to read some literature to read this work. I suggest the authors could add a little bit more details on the deduction of Huber loss and its current applications on bandits and reinforcement learning problems. For example, it would be better if the authors could explain some details of [Sun et al. 2020]. I notice the work [Kang & Kim 2023] also used the same loss Huber regression for heavy-tailed bandit, so the authors should highlight the similarity and differences between Kang & Kim 2023 and yours, which is missing now. I also find [1] and [2] about Huber regression in bandits/RL, and the authors should also discuss these two works in the revision: [1] studies the matrix linear contextual bandit with Huber regression for heavy-tailed rewards, which is an extension to SLB and obtains similar regret bound. [2] considers Huber regression in distributed reinforcement learning, which also considers a similar framework in RL parameter adaptation.


[1] Kang et al. Low-rank Matrix Bandits with Heavy-tailed Rewards

[2] Malekzadeh et al. A Robust Quantile Huber Loss With Interpretable Parameter Adjustment In Distributional Reinforcement Learning

**Experimental Designs Or Analyses:**

1. It seems that the experiments only consider the case when the noise has fixed $\nu_t$, which may be too weak. It would be better if the authors could show your algorithm can yield robust performance under varying $\nu_t$ across time $t$.
2. It seems that several existing methods such as CRTM, CRMM and HeavyOFUL can yield competitive or even better empirical performance with less computational cost. This raises doubts about the advantage of the proposed approach.

**Methods And Evaluation Criteria:**

The experimental setting and evaluation criteria are aligned with the state-of-the-art evaluation framework in bandit.

**Other Comments Or Suggestions:**

I think this paper has its own merit. I will reconsider my rating on this work during the rebuttal phase, and potentially raise or decrease my rating based on the discussion with the authors and other reviewers.

**Other Strengths And Weaknesses:**

Some typos: 1. line 39 right part: "cannot not". 2. line 231 left part: "both of" should be changed to "both". 3. line 443: "none of" instead of "none". 4. The estimator in line 193 left part: there are $\hat\theta_{t+1}$ in both two equations, which is a little bit confusing. Although I can finally get it, but it is better to change $\hat\theta_{t+1}$ to something else in the first equations. 5. There are some non-English words in the submitted codes in the supplementary files, which are not very professional and may lead to the violation of ICML regulations.

**Questions For Authors:**

1. Can the approach generalize to the generalized linear bandit setting, or is it strictly limited to the linear case?
2. How sensitive is the proposed method to hyperparameter tuning (e.g., Huber threshold selection)? I do not see an ablation study on parameters or the hyperparameter setting in the manuscript.
3. I feel the regret curves for most algorithms in your experiments are linear instead of sublinear, which looks contradicts with the theoretical results. Can you explain that?
4. In [Kang & Kim et al.] and [Kang et al. Low-rank Matrix Bandits with Heavy-tailed Rewards] (mentioned above), the Huber loss function does not contain the variance normalization factor $\sigma_s$. Is that because they assume the variance is fixed? Therefore, if the upper bound $\nu$ is known, then your Huber loss can be free of the normalization factor as well?

**Relation To Broader Scientific Literature:**

This paper integrates adaptive robust estimation (Huber regression), computationally efficient online learning (OMD), and variance-aware regret analysis into a single unified framework. As the authors point out in Section 5, the proposed methodology of this work has potential in other problems such as linear MDP problem and online adaptive control problem.

**Theoretical Claims:**

I think the theoretical claims are correct. I will look into other reviewers' opinions for further reference. However, the current method can not handle the case when $\nu_t$ is unknown, which weakens the theoretical contributions of this work.

---

> ### Author Rebuttal · Authors · 2025-03-31
>
> Thanks for your helpful comments. Below, we address your main questions regarding the technical contributions, extensions to GLB, experiments, and related works. For other minor issues (e.g., typos), due to limited space, we will directly revise the paper according to your suggestions.
>
> ---
>
> **Q1.** "cannot handle the case when $\nu_t$ is unknown, which weakens the theoretical contributions"
>
> **A1.** Thanks for the comment. We'd like to remind that knowing the moment $\nu_t$ is shared by our work and previous works for heavy-tailed linear bandits, and this issue is orthogonal to our main contributions.
>
> - **Relying on moment $\nu_t$**: Note that existing works including the current SOTA [Huang et al., 2024] require the moment information $\nu_t$, even if they can use an offline MLE estimator. Similarly, our work also faces this issue; however, in contrast, we can ensure one-pass efficiency. Handling unknown $\nu_t$ is indeed important and challenging for the community, which is left as future work to explore.
> - **Theoretical contribution**: Even with known $\nu_t$, designing a one-pass algorithm remains highly non-trivial due to several challenges, including using one-pass OMD to approximate full-batch MLE, and handling Huber loss and heavy-tailed noise. We kindly request the reviewer to refer to **A2 for Reviewer McYB** for more details. Thanks!
>
> ---
>
> **Q2.** "...generalize to the generalized linear bandit (GLB)..."
>
> **A2.** In fact, there is prior work that applies OMD to logistic bandits (an important class of GLB) with sub-Gaussian noise, while our paper uses OMD for linear bandits with heavy-tailed noise. This suggests a potential combination. Nonetheless, an evident challenge is how to design a suitable Huber-type regression loss tailored to nonlinear link functions of GLB for the offline MLE estimator. We leave this as an exciting direction for future work. Thanks!
>
> ---
>
> **Q3.** Experiments on several aspects ("time-varying $\nu_t$"; "the empirical advantage of HvtLB"; "parameter sensitivity"; "linear regret curves", etc)
>
> **A3.** Thanks for those careful comments! To address your concerns, we have conducted additional experiments and comparisons in the anonymous URL https://default-anno-bucket.s3.us-west-1.amazonaws.com/HvtLB_revised_exp.pdf
>
> Including, varying $\nu_t$ in Figure 2, empirical advantage in Figure 4, and linear regret issue in Figure 5. For parameter sensitivity,  key parameters in the algorithm, such as Huber threshold, are not chosen by tuning but are derived from theoretical analysis. As a result, they indeed can be relatively sensitive. We will add these results in the revised version.
>
> ---
>
> **Q4.** More details on [Sun et al. 2020], and differences between [Kang & Kim]
>
> **A4.** Thanks for the helpful suggestion. In the revised version, we will include more details on [Sun et al. 2020]. As for [Kang & Kim], the similarity in titles may cause confusion. However, there are key differences in both setting and methods.
>
> - **Setting**: We focuses on linear bandits with **infinite arm set**, where context $X_t$ chosen depends on previous $\\{X_s\\}_{s<t}$. Our goal is to design a one-pass algorithm with optimal regret. In contrast, [Kang & Kim] studies linear contextual bandits with **finite arm set**, where $X_t$ is **i.i.d. sampled**. Their method is based on offline MLE and is *not* online.
>
> - **Algorithm**: Even with a similar Huber regression, the designs differ fundamentally. We use OMD with a carefully designed recursive normalization factor, while [Kang & Kim] uses offline MLE with a forced exploration mechanism. Please refer to **A5** for more details.
>
> ---
>
> **Q5.** "In [Kang & Kim] and [Kang et al.], ... if the upper bound $\nu$ is known, then your Huber loss can be free of the normalization factor as well?"
>
> **A5.** Thanks for the question. Our Huber loss still requires this factor, even if $\nu$ is known. The normalization factor $\sigma_s$ not only ensures variance-awareness but also guarantees that the denoised data lies within the quadratic region of the Huber loss. This ensures that the Hessian has a lower bound, which is crucial for the analysis. We achieve this by designing a recursive factor, where $\sigma_t$ depends on the last round confidence bound $\beta_{t-1}$ (see line 5 in Algorithm 1).
>
> While these two works avoid this factor, they introduce additional algorithmic mechanisms and assumptions to achieve a similar goal: (i) [Kang & Kim] introduces a forced exploration strategy and Assumption 4 to ensure that the Hessian has a lower-bounded minimum eigenvalue; (ii) [Kang et al.] adopts the local adaptive majorize-minimization method and Assumption 3.1 to establish the Local Restricted Strong Convexity condition (Definition A.3), which guarantees a lower bound on the Hessian.
>
> ---
>
> We are grateful for your careful review! If our clarifications have adequately addressed your concerns, please consider updating your score. Thanks!

---

> > ### Comment · Reviewer_P7Rp · 2025-04-06
> >
> > Thank you for your responses. After reading the authors' rebuttal, I feel this work has its merit and is an interesting exploration on heavy-tailed linear bandits. Could you please include the arguments about the difficulty of agnostic to $\epsilon$ and $\nu_t$, and the detailed comparison with Sun et al. 2020, Kang & Kim 2023 and Kang et al. 2024 (responses to my Q4 and Q5) into the next revision? I think that is very helpful for readers like me who are not familiar with Huber regression.

---

> > > ### Author Response · Authors · 2025-04-06
> > >
> > > Thank you once again for your thoughtful review and for appreciating our work! We will certainly  incorporate these discussions (**A4** and **A5**) into the revised version and improve the presentation based on your suggestions to make the paper clearer and easier to understand.
> > >
> > > Moreover, we thank the reviewer for pointing out the related works [Kang et al., 2024] and [Malekzadeh et al., 2024], which were previously missing from our paper discussion. After a careful check recently, we found that [Kang et al., 2024] is highly relevant to our work in terms of both the problem setting and methodology.
> > > - Problem setting: both works consider the heavy-tailed nosie with finite $(1+\epsilon)$-moment. The main difference lies in the reward model: our work considers vector-valued context and parameter, i.e., $X_t, \theta_* \in\mathbb{R}^d$, while theirs considers matrix-valued context and parameter, i.e., $X_t, \theta_* \in\mathbb{R}^{d_1\times d_2}$.
> > > - Methodology: both works adopt the Huber loss. Our approach adopts a one-pass update scheme, while theirs relies on offline MLE. As discussed in **A5**, both methods incorporate additional algorithmic designs aimed at the same goal: ensuring a lower-bounded Hessian. This may suggest that our approach may have the potential to enable one-pass updates in their setting as well.
> > >
> > > We will include a discussion of these works in the revised version.
> > >
> > > Finally, we sincerely appreciate your recognition of our technical contributions. As we will continue to refine the presentation and further enrich the discussion of related works, we would be deeply grateful if you could kindly consider raising your score to further support our paper. Thank you again for your valuable feedback!

---

### Official Review · Reviewer_ZqBj · 2025-03-10

**Overall Recommendation:** 2

**Summary:**

This work introduces a new algorithm, inspired by [1], where solve the computation issues with the algorithms in [1].

**Claims And Evidence:**

see Other Strengths And Weaknesses

**Essential References Not Discussed:**

see Other Strengths And Weaknesses

**Experimental Designs Or Analyses:**

see Other Strengths And Weaknesses

**Methods And Evaluation Criteria:**

see Other Strengths And Weaknesses

**Other Comments Or Suggestions:**

see Other Strengths And Weaknesses

**Other Strengths And Weaknesses:**

Weaknesses:

1- In Eq. (7), replace the notation with $ z_t(\theta) $.

2- Compared to [1], the contribution of this work is incremental.

3- It would be beneficial to elaborate on the main theoretical contributions in relation to [1].

4- In the experiments, for a Student’s t-distribution with a degree of freedom greater than 2, the second moment is bounded. It would be valuable to explore the case where the degree of freedom satisfies $ 1 < df \leq 2 $.

5- The current experiments are insufficient, as they are conducted solely on synthetic datasets.

6- The present definition of heavy-tailed distributions does not encompass the sub-Gaussian case. Therefore, I suggest discussing only the heavy-tailed case in Section 4 to avoid potential confusion.

----

References:

[1]: Huang, J., Zhong, H., Wang, L., and Yang, L. Tackling heavy-tailed rewards in reinforcement learning with function approximation: Minimax optimal and instance-dependent regret bounds. NIPS, 2024.

**Questions For Authors:**

see Other Strengths And Weaknesses

**Relation To Broader Scientific Literature:**

see Other Strengths And Weaknesses

**Theoretical Claims:**

see Other Strengths And Weaknesses

---

> ### Author Rebuttal · Authors · 2025-03-31
>
> Thanks for your helpful suggestions. Below we will address your main concerns regarding the technical contributions and report additional experiments.
>
> ---
>
> **Q1.** about the contributions ("Compared to [1], the contribution of this work is incremental", "elaborate on the main theoretical contributions in relation to [1]")
>
> **A1.** Thank you for the comments. We believe there may have been a misunderstanding due to our insufficient emphasis on the technical contributions, particularly the challenges of applying OMD to heavy-tailed bandits. We will revise the paper to ensure this aspect is presented more clearly. Here, we'd like to take this opportunity to clarify these challenges and further emphasize our contributions.
>
> - **Challenge 1: Using one-pass OMD to approximate full-batch MLE.** The offline MLE estimator leverages the entire history of data, enabling its estimation error to be well-controlled. In contrast, OMD updates the one-pass estimator using only current data, making it inherently challenging to ensure the final regret remains unaffected. While OMD is known for its effectiveness in minimizing regret, adapting this capability into a good statistical estimator is far from straightforward. In fact, as shown in Section 4.1, even under Sub-Gaussian noise (a significantly easier scenario), deploying OMD requires: (i) carefully designing the local norm to collaborate with self-normalized concentration, and (ii) properly using the negative term in the regret analysis to control the generalization gap.
> - **Challenge 2: Handing Huber loss and heavy-tailed noise.** In the heavy-tailed setting, deploying OMD as a one-pass estimator is even more challenging. The primary difficulty arises from the curvature of the Huber loss, which is partially linear (undesired region) and partially quadratic (desired region), necessitating careful control of the threshold. Furthermore, in the estimation error analysis, the bias introduced by the one-pass OMD update manifests as an additional stability term that can grow as $\mathcal{O}(\sqrt{T})$. It is critical to carefully design both the adaptive normalization factor of the Huber loss and the learning rate in OMD, together to effectively manage the estimation error.
>
> We will revise the paper to ensure these challenges and our technical contributions are clearly elaborated.
>
> ---
>
> **Q2.** About experiment  "explore the case where the degree of freedom satisfies $1<df\leq 2$"
>
> **A2.** Thank you for the suggestions. We have conducted additional experiments to address your concerns. We summarize all the details and results in the following anonymous URL: https://default-anno-bucket.s3.us-west-1.amazonaws.com/HvtLB_revised_exp.pdf
>
> Please refer to Figure 2 in this anonymous URL for details, where we use $df = 1.7$ for the Student’s t-distributions. Additionally, we also tested other heavy-tailed noise distributions and ensured they satisfy the condition that there exists an $\epsilon \in (0, 1)$ such that the $(1+\epsilon)$-th moment is bounded, which can be found in Figure 1. We will add these results in the revised version.
>
> ---
>
> **Q3.** About experiment "insufficient, as they are conducted solely on synthetic datasets"
>
> **A3.** Thanks for your comments. We'd like to emphasize that our paper primarily focuses on the theoretical side, which we have elaborated the technical contributions in detail (please refer to **A1**). The synthetic data are well-suited for setting different configurations to support our theoretical findings. On the practical side, it might be more interesting to explore online MDPs and adaptive control, given the fundamental connection between linear bandits and decision-making problems. We believe that our proposed algorithm (with suitable modifications) could be highly useful. We leave these extensions for future investigation and prefer not to spend more spaces for them in this paper to avoid diverting readers from the main focus.
>
> ---
>
> **Q4.** "In Eq.(7), replace the notation with $z_t(\theta)$"
>
> **A4.** Thank you for pointing out the confusion, we will make the presentation clearer in the revised version.
>
> ---
>
> **Q5.** "heavy-tailed distributions does not encompass the sub-Gaussian case...I suggest discussing only the heavy-tailed case in Section 4 to avoid potential confusion"
>
> **A5.** Thank you for this important suggestion. We intended to use the sub-Gaussian case as a simplified example to demonstrate how to deploy the OMD estimator in linear bandits. However, your suggestion is very reasonable, and we will revise the paper to avoid this confusion.
>
> ---
>
> Thanks for your helpful suggestions. We will add more experiments and improve the paper writing to more clearly emphasize the technical contributions. If our responses have properly addressed your concerns, please consider updating your score. Thanks!

---

> > ### Comment · Reviewer_ZqBj · 2025-04-02
> >
> > I appreciate the authors' efforts in the rebuttal.
> > Conducting experiments on real datasets is crucial.
> >
> > Is it possible to apply other loss function for this scenario? For example check tilted risk minimization [1].
> >
> > [1]: Li, Tian, et al. "Tilted empirical risk minimization." arXiv preprint arXiv:2007.01162 (2020).

---

> > > ### Author Response · Authors · 2025-04-04
> > >
> > > Thanks for your appreciation of our rebuttal and suggestions for the real dataset! We plan to incorporate more experiments on real datasets in the next version based on your suggestions.
> > >
> > > **Regarding the possibility of other loss functions.** Thank you for the insightful question and the provided reference. Both tilted empirical risk minimization (TERM) [1] and Huber loss aim to mitigate the impact of outlier losses during empirical risk minimization (ERM). While Huber loss is specifically designed for the squared loss, TERM provides a more general weighting mechanism applicable to a wide range of loss functions. This suggests that TERM can effectively handle offline linear regression under heavy-tailed noise and has the potential to achieve sound theoretical guarantees, as evidenced by its empirical advantage over Huber loss (see Section 5.1 of [1]).
> > >
> > > ***However, in the online learning scenario, from our current understanding, TERM may lack sufficient flexibility to support one-pass update as effectively as Huber loss.***
> > >
> > > - Huber loss defines its penalty locally: the loss of each new sample can be independently determined to follow either a quadratic or linear regime. Adding a new data point does not affect the penalty of previous samples, making Huber loss particularly well-suited for the online scenario.
> > > - TERM, on the other hand, relies on a log-sum-exp aggregation over *all* sample losses, i.e., $\frac{1}{t} \log (\frac{1}{N} \sum_{i=1}^N e^{t f(x_i ; \theta)})$. This means that the weight of each sample is determined relative to the rest of the dataset. Adding a new sample alters the overall distribution of losses and changes the relative weighting across all data points. This global coupling implies the re-computation of the entire objective for each new sample, making TERM potentially unsuitable for one-pass updates in online learning, or at least its extension to online learning is non-trivial.
> > >
> > > We will include a discussion of this alternative loss function for handling outliers [1] in the revised version.
> > >
> > > If our clarifications have properly addressed your concerns, please consider updating your score. Thanks!

---

### Official Review · Reviewer_kVUc · 2025-03-13

**Overall Recommendation:** 4

**Summary:**

This paper studies the heavy-tailed linear bandits problem. Doing OMD on the Huber loss, the authors yielded an algorithm with near-optimal $\tilde{\mathcal O}(d T^{1/(1+\epsilon)})$ regret which only needs $\mathcal O(1)$ computation per round (instead of doing a huge Huber regression as previous ones in the literature). Furthermore, the algorithm also attains a "variance-aware" property that automatically scales with the $(1+\epsilon)$-th moments.

**Claims And Evidence:**

The theorems are complemented with easy-to-follow sketches.

**Essential References Not Discussed:**

Looks good. See questions for some not-so-related ones.

**Experimental Designs Or Analyses:**

The numerical illustration is valid.

**Methods And Evaluation Criteria:**

The setup and notation is standard and consistent with previous works in the literature.

**Other Comments Or Suggestions:**

No

**Other Strengths And Weaknesses:**

Strength:

1. The writing is pretty clear, introducing previous ideas on heavy-tailed linear bandits and highlighting the technical difference / innovation.
2. The analysis sketch is well-written, making it easy to follow.

Weakness:

1. The algorithm requires knowledge of $\epsilon$ and $\nu_t$. I feel it's acceptable as it's the common challenge for heavy-tailed linear bandits (see Q1 & Q2).

**Questions For Authors:**

1. Looks like all previous heavy-tail linear bandit algorithms require the exact knowledge of $\epsilon$ and $\nu_t$. Recently this has been relaxed in the heavy-tail multi-armed bandit literature: [1] for the adversarial case, [2] for the stochastic case, and [3] for both ("best-of-both-worlds"). I know the techniques here and there are substancially different, but it'd be great to still give some discussions.
2. Also, regarding the knowledge of $\nu_t$, there has been Empirical Bernstein inequalities which removes this for standard linear bandits [4, 5]. Is there any similar self-bounding concentrations in the world of heavy-tailed distributions? If yes, could you briefly explain what's the main issue when doing it for the learning setup?

[1] Jiatai Huang, Yan Dai, and Longbo Huang. "Adaptive best-of-both-worlds algorithm for heavy-tailed multi-armed bandits." ICML 2022.

[2] Gianmarco Genalti, Lupo Marsigli, Nicola Gatti, and Alberto Maria Metelli. "(ε,𝑢)-Adaptive Regret Minimization in Heavy-Tailed Bandits." COLT 2024.

[3] Yu Chen, Jiatai Huang, Yan Dai, and Longbo Huang. "uniINF: Best-of-Both-Worlds Algorithm for Parameter-Free Heavy-Tailed MABs." ICLR 2025.

[4] Zihan Zhang, Jiaqi Yang, Xiangyang Ji, and Simon S Du. "Improved variance-aware confidence sets for linear bandits and linear mixture mdp." NeurIPS 2021.

[5] Yeoneung Kim, Insoon Yang, and Kwang-Sung Jun. "Improved regret analysis for variance-adaptive linear bandits and horizon-free linear mixture mdps." NeurIPS 2022.

**Relation To Broader Scientific Literature:**

Can be interesting to other problems with heavy-tailed distributions; but mostly talored towards the heavy-tailed linear bandits

**Theoretical Claims:**

I didn't check the details of the proof, but the sketches are convincing enough.

---

> ### Author Rebuttal · Authors · 2025-03-31
>
> Thanks for the valuable feedback and appreciation of our work! We will revise the paper accordingly. Below, we answer your questions.
>
> ---
>
> **Q1.** More discussions about algorithms without knowledge of $\epsilon$ and $\nu_{t}$ in advance.
>
> **A1.** Thanks for the suggestion. Relaxing this assumption is indeed an important direction for future research of heavy-tailed bandits. The literature and techniques you mentioned for removing the need for known $\nu_t$ could potentially be adapted to the heavy-tailed setting. We will include a more detailed discussion of these works in the revised version.
>
> ---
>
> **Q2.** " Is there any similar self-bounding concentrations in the world of heavy-tailed distributions? If yes, could you briefly explain what's the main issue when doing it for the learning setup?"
>
> **A2.** Thanks for the question. As far as we know, Existing results based on empirical variance typically require either a bounded norm or sub-Gaussian assumption which is not applicable to heavy-tailed distributions, since in the heavy-tailed setting (e.g., when only the $(1+\epsilon)$-moment is bounded), the empirical variance itself can be unbounded.
>
> Besides, after our submission, we noticed a recent paper published on arXiv that studies the heavy-tailed bandit setting (with $\epsilon = 1$) and claims to eliminate the need for knowing the variance using a peeling-based algorithm, under certain assumptions [1]. Although we have not studied the paper in detail yet, we believe its result is noteworthy.
>
> [1] Ye et al., Catoni Contextual Bandits are Robust to Heavy-tailed Rewards.
>
> ---
>
> We are grateful for your careful review! We will provide more discussion about the literature you suggested in the revised version.

---

> > ### Comment · Reviewer_kVUc · 2025-04-01
> >
> > Thank you. I like this paper overall. I encourage the authors to incorporate above discussions into Related Works or Conclusions in the revision.

---

> > > ### Author Response · Authors · 2025-04-04
> > >
> > > Thanks for your appreciation of our work! We will incorporate these discussions into the revised version.

---

### Official Review · Reviewer_C6wo · 2025-03-20

**Overall Recommendation:** 3

**Summary:**

The paper proposes a method for linear bandits with heavy-tailed noise variable, that efficiently estimates the bandit parameter with a single pass through the data and doesn't require processing the complete historical data.

**Claims And Evidence:**

It is claimed that the method can be adapted to more generalized decision-making scenarios, such as reinforcement learning, but no theoretical or experimental evidence is provided, even though a discussion is provided.

**Essential References Not Discussed:**

The literature of the offline bandit learning could be explored more. Especially off-policy evaluation, where MOM and truncation methods are provided to tackle heavy-tailed noise and outlier effect. For example, the following papers could be studied and reviewed.

Sakhi, Otmane, et al. "Logarithmic smoothing for pessimistic off-policy evaluation, selection and learning." arXiv preprint arXiv:2405.14335 (2024).

Behnamnia, Armin, et al. "Batch Learning via Log-Sum-Exponential Estimator from Logged Bandit Feedback." ICML 2024 Workshop: Aligning Reinforcement Learning Experimentalists and Theorists.

**Experimental Designs Or Analyses:**

The. experiment setup is very limited and insufficient to prove the claims of the authors.
1. One of the claims of the paper was that previous studies assumed a fixed action set, while the proposed method doesn't require such an assumption. But in the experiments, the action set is a fixed set of 50 arms.
2. The same statement can be mentioned. for the case of Reinforcement Learning or other decision-making scenarios.
3. There is a vast range of heavy-tailed distributions, but only student-t distribution is tested in the experiments, which is specifically very similar to normal distribution, hence the good performance on only the t distribution is not enough.

**Methods And Evaluation Criteria:**

The evaluation criteria are standard, as regret bound and time complexity are the common metrics in the literature.

**Other Comments Or Suggestions:**

I don't have any particular suggestion.

**Other Strengths And Weaknesses:**

The contributions of the paper are weak. Altogether it provides a theoretical analysis of huber-based linear bandit parameter estimation with OMD optimization, with insufficient experimental evidence.

**Questions For Authors:**

The paper's structure is clear. I don't have any question.

**Relation To Broader Scientific Literature:**

The contribution of the paper is not significant and essential. Compared to the Heavy-OFUL, whichi is a strong SOTA in the literature, the only contribution is the replacement of the batch GD with OMD algorithm which allows an online per-sample update. The idea of Huber loss and its application on heavy-tailed linear bandits have been thoroughly discussed in Heavy-OFUL and the OMD algorithm is a well-known optimization algorithm. Yet, the theoretical analysis of the combination of OMD with huber objective can be views as the only contribution of the paper.

**Theoretical Claims:**

The theoretical claims are built towards a single, important theorem: the regret bound. The proof sketch provided in section 4 make the proof process easier to follow. The major part of the theoretical stuff focuses on the estimation error of the bandit parameter, which is essential for a regret analysis.

---

> ### Author Rebuttal · Authors · 2025-03-31
>
> Thanks for your suggestions on experiments and literatures. We will revise the paper accordingly. However, there is an important misunderstanding regarding our technical contributions that we need to clarify.
>
> ---
>
> **Q1.** "contribution is not significant...only contribution is the replacement of the batch GD with OMD..."
>
> **A1:** We respectfully disagree with the comments. The SOTA method (HEAVY-OFUL) [Huang et al., 2024] adopts an offline MLE estimator and achieves optimal regret for heavy-tailed linear bandits. However, ensuring the one-pass" property is crucial for online algorithms to ***update efficiently without storing the entire historical data***. Our work builds on HEAVY-OFUL and develops an OMD-based one-pass algorithm, while also ensuring that the final regret remains unaffected. **This is technically highly non-trivial.** We believe it is unfair to describe as merely "replacing the batch GD with the OMD algorithm for online per-sample updates". Below, we outline the technical challenges in more detail.
>
> - **Challenge 1: Using one-pass OMD to approximate full-batch MLE.** The offline MLE estimator leverages the entire history of data, enabling its estimation error to be well-controlled. In contrast, it is intuitively non-trivial to develop a one-pass estimator (which uses only the current data) while still ensuring that the estimation error remains as good as the offline MLE estimator. In fact, as shown in Section 4.1, even under Sub-Gaussian noise (a significantly easier scenario), deploying OMD requires: (i) carefully designing the local norm to collaborate with self-normalized concentration, and (ii) properly using the negative term in the regret analysis to control the generalization gap.
> - **Challenge 2: Handing Huber loss and heavy-tailed noise.** In the heavy-tailed setting, deploying OMD as a one-pass estimator is even more challenging. The primary difficulty arises from the curvature of the Huber loss, which is partially linear (undesired region) and partially quadratic (desired region), necessitating careful control of the threshold. Furthermore, in the estimation error analysis, the bias introduced by the one-pass OMD update manifests as an additional stability term that can grow as $\mathcal{O}(\sqrt{T})$. It is critical to carefully design both the adaptive normalization factor of the Huber loss and the learning rate in OMD, together to effectively manage the estimation error.
>
> Overall, we believe these technical contributions are not only interesting to the bandits community but also have a broad impact on the audiences of online MDPs and online adaptive control. We will improve the paper's writing to emphasize these points more clearly. Thanks!
>
> ---
>
> **Q2.** "offline bandit learning could be explored more"
>
> **A2.** Thanks for pointing out the literature on offline bandit learning, which relates to managing extreme values caused by importance weighting. We are happy to incorporate a discussion in the revision.
>
> ---
>
> **Q3.** "Experiment setup is very limited and insufficient to prove the claims"
>
> **A3.** We'd like to emphasize that our paper primarily focuses on the theoretical side, which we have elaborated the technical contributions  in detail (please refer to **A1**).  Experiments are intended to support our theoretical findings.
>
> We are happy to conduct additional tests to better address your concerns, and new results and plots are summarized in the following anonymous URL: https://default-anno-bucket.s3.us-west-1.amazonaws.com/HvtLB_revised_exp.pdf
>
> Concretely, we add experiments of various heavy-tailed noise (including Pareto, Lomax, Fisk); time-varying $\nu_t$; varying arm set case; more comparation about empirical advantage of our methods. We will include these additional tests in the revised version. Hope these will address your concerns.
>
> ---
>
> **Q4.** "It is claimed that the method can be adapted to more generalized decision-making ... but no theoretical or experimental evidence is provided"
>
> **A4.** Without a doubt, the linear bandit model is fundamental in online learning. Our work makes an important step for linear bandits with heavy-tailed noise. As mentioned, our technical contributions are already substantial enough to gain interest from the community.
>
> Given the importance of linear bandits, the discussion in Section 5 is intended to highlight the potential of our Hvt-LB algorithm to broader decision-making scenarios. However, we prefer not to spend more spaces for those extensions to avoid diverting readers from the main focus, i.e., linear bandits. We believe our techniques will be of interest to audiences working on online linear MDPs and adaptive control.
>
> ---
>
> We appreciate your suggestions and will add additional experiments and related works accordingly. We will also emphasize technical contributions more clearly. If our clarifications have adequately addressed your concerns, please consider updating your score. Thanks!

---

> > ### Comment · Reviewer_C6wo · 2025-04-07
> >
> > I thank the authors for the additional experiments.
> > I appreciate the authors' work on the design of an OMD algorithm based on Huber loss. The algorithm altogether is indeed novel, and hence the theoretical analysis is so. I can summarize the contribution as proposing an online method that can do as well as the well-known offline Heavy-OFUL method, in the presence of heavy-tailed rewards (which is part of the contribution of Heavy-OFUL itself).
> > Are there any experimental designs for which only the one-step historyless approaches are computationally feasible?

---

> > > ### Author Response · Authors · 2025-04-07
> > >
> > > Thank you for your recognition of the novelty and technical contributions of our work! Your summary accurately captures the essence of our contribution: designing a novel online algorithm (which relies only on current data) to achieve an optimal regret guarantee same as the MLE-based offline algorithm (which uses the entire historical data).
> > >
> > > **Regarding the question about experimental settings with computational feasibility:** a feasible online algorithm should have a per-round update complexity independent of the iteration count (i.e., historyless). Otherwise, as the online data stream grows, the per-round computational cost will increase, eventually making the update infeasible (since it requires using the entire historical data).
> > >
> > > To provide a concrete example: completing an 18,000-round online estimation takes **approximately 3 hours** with Heavy-OFUL, whereas our proposed one-pass method (and other one-pass baselines) finishes in under **1 minute** (as shown in Figure 1&2 in paper). This significant difference becomes more problematic when running repeated experiments or tuning hyperparameters.
> > >
> > > The computational infeasibility of the offline algorithm becomes even more severe in realistic applications, where online estimation often requires timely updates that may lose their value if delayed. We plan to include experiments on real data in a future version to better illustrate the practical limitations of MLE-based offline approaches, according to your suggestion.
> > >
> > > ---
> > >
> > > Finally, we sincerely appreciate your recognition of our contributions. As we continue to add more experiments and discussions on related works, we would be deeply grateful if you would consider raising your score to further support our submission. Thank you again for your valuable feedback!

---

### Official Review · Reviewer_McYB · 2025-03-24

**Overall Recommendation:** 3

**Summary:**

This paper proposes a novel, computationally efficient algorithm for heavy-tailed linear bandits that achieves the best-known regret bound in the literature.

**Claims And Evidence:**

The claims are clear and supported by convincing evidence from the literature.

**Essential References Not Discussed:**

All the essential related references are discussed in the paper.

**Experimental Designs Or Analyses:**

While the experimental design is standard, I suggest the authors provide results for varying $\nu_t$ to support the claim that the algorithm is adaptive to $\nu_t$

**Methods And Evaluation Criteria:**

The methods and evaluation criteria follow standards in the literature.

**Other Comments Or Suggestions:**

In Lemma 1, $\sigma_{\min}$ is used without being defined beforehand.



============== After Rebuttal ================

I appreciate the authors detailed responses.
I acknowledge that this work has novel contribution on heavy-tailed bandits.
Please highlight the responses in the revision for clarity, especially on the novelty.

**Other Strengths And Weaknesses:**

Strengths: The paper is well-organized, with a thorough discussion that is easy to follow as well as with clear motivation.

Weaknesses: The novelty seems limited, as the challenges of combining online mirror descent with heavy-tailed bandits are not discussed clearly.

**Questions For Authors:**

Q1: How will the empirical performance of the algorithm vary with changing $v_t$?

Q2: Is there a way to remove the requirement for knowledge of the horizon time $T$?

Q3: Could the authors provide specific challenges in deriving the results of applying online mirror descent to heavy-tailed bandits?

**Relation To Broader Scientific Literature:**

As the authors mentioned in the discussion section, their results are applicable to linear Markov decision processes and online adaptive control. Because online estimation for heavy-tailed distributions is not well established in bandits or reinforcement learning, I believe this work will have a wide impact on those literatures

**Theoretical Claims:**

The proofs are based on standard techniques for heavy-tailed distributions and online mirror descent

---

> ### Author Rebuttal · Authors · 2025-03-31
>
> Thanks for your careful review! In the following, we will address your main concerns and report additional experiments.
>
> ---
>
> **Q1.** "How will the empirical performance of the algorithm vary with changing $\nu_t$? I suggest the authors provide results for varying $\nu_t$ to support the claim that the algorithm is adaptive to $\nu_t$."
>
> **A1.** Thanks for the suggestion. We have conducted additional experiments of time-varying $\nu_t$ to further support the variance-aware property of our method. We summarize all the details and results in the following anonymous URL: https://default-anno-bucket.s3.us-west-1.amazonaws.com/HvtLB_revised_exp.pdf
>
> Please refer to Figure 2 of this anonymous URL for details. We will add these results in the revised version.
>
> ---
>
> **Q2.** " The novelty seems limited, as the challenges of combining online mirror descent with heavy-tailed bandits are not discussed clearly. Could the authors provide specific challenges in deriving the results of applying online mirror descent to heavy-tailed bandits?"
>
> **A2.** Thank you for the comments. We believe there may have been a misunderstanding due to our insufficient emphasis on the challenges of applying OMD to heavy-tailed bandits. We will revise the paper to ensure this aspect is presented more clearly. Here, we'd like to take this opportunity to clarify these challenges.
>
> - **Challenge 1: Using one-pass OMD to approximate full-batch MLE.** The offline MLE estimator leverages the entire history of data, enabling its estimation error to be well-controlled. In contrast, OMD updates the one-pass estimator using only current data, making it inherently challenging to ensure the final regret remains unaffected. While OMD is known for its effectiveness in minimizing regret, adapting this capability into a good statistical estimator is far from straightforward. In fact, as shown in Section 4.1, even under Sub-Gaussian noise (a significantly easier scenario), deploying OMD requires: (i) carefully designing the local norm to collaborate with self-normalized concentration, and (ii) properly using the negative term in the regret analysis to control the generalization gap.
> - **Challenge 2: Handing Huber loss and heavy-tailed noise.** In the heavy-tailed setting, deploying OMD as a one-pass estimator is even more challenging. The primary difficulty arises from the curvature of the Huber loss, which is partially linear (undesired region) and partially quadratic (desired region), necessitating careful control of the threshold. Furthermore, in the estimation error analysis, the bias introduced by the one-pass OMD update manifests as an additional stability term that can grow as $\mathcal{O}(\sqrt{T})$. It is critical to carefully design both the adaptive normalization factor of the Huber loss and the learning rate in OMD, together to effectively manage the estimation error.
>
> We will revise the paper to ensure these challenges are clearly elaborated.
>
> ---
>
> **Q3.** "In Lemma $1$, $\sigma_{\min }$ is used without being defined beforehand."
>
> **A3.** Thank you for the detailed review! We will correct this in the revised version, specifically, $\sigma_{\min }$ is a small positive constant to avoid singularity, and we provide the setting of $\sigma_{\min}$ in Theorem 1.
>
> ---
>
> **Q4.** "Is there a way to remove the requirement for knowledge of the horizon time $T$?"
>
> **A4.** Thanks for the question. The requirement of a known time horizon $T$ can be removed using the doubling trick. Specifically, we let the algorithm restart at time steps $2^1, 2^2, 2^3, 2^4, \ldots, 2^M$. If the time horizon is $T$, then the total number of intervals is $M \approx \log T$. When the time horizon is known, the regret bound is $\widetilde{\mathcal{O}}(T^{\frac{1}{1+\epsilon}})$. Since the length of each interval $2^m$ is known in advance, we can apply the same bound within each interval. Then the overall regret can be calculated by:
>
> $$\widetilde{\mathcal{O}}\left(\sum_{m=1}^M\left(2^m\right)^{\frac{1}{1+\epsilon}}\right)=\widetilde{\mathcal{O}}\left(\sum_{m=1}^M 2^{\frac{m}{1+\epsilon}}\right)=\widetilde{\mathcal{O}}\left(2^{\frac{1}{1+\epsilon}} \frac{2^{\frac{M}{1+\epsilon}}-1}{2^{\frac{1}{1+\epsilon}}-1}\right)=\widetilde{\mathcal{O}}\left(\frac{2^{\frac{1}{1+\epsilon}}}{2^{\frac{1}{1+\epsilon}}-1}\left(T^{\frac{1}{1+\epsilon}}-1\right)\right) = \widetilde{\mathcal{O}}(T^{\frac{1}{1+\epsilon}}).$$
>
> In this way, we are able to remove the requirement of known $T$ in advance, while still achieving a regret bound of $\widetilde{\mathcal{O}}(T^{\frac{1}{1+\epsilon}})$, up to a constant factor overhead.
>
> ---
>
> Thanks again for your insightful comments. We will add more experiments and improve the presentation in the revised version.

---

> > ### Comment · Reviewer_McYB · 2025-04-08
> >
> > I appreciate the authors comments and efforts for the additional experiments.
> >
> > On Question 2, could the authors point out the novel technical analysis or results that overcomes the mentioned challenges?

---

> > > ### Author Response · Authors · 2025-04-09
> > >
> > > Thanks again for your careful review and your appreciation of our efforts! In the following, we explain how we address the two mentioned challenges: one arising from one-pass update and the other from the heavy-tailed noise. Specifically,
> > >
> > > **(i) One-pass Update.** In the estimation error analysis, the one-pass update introduces the additional stability term compared to offline algorithms, specifically, $\sum\_{s=1}^t \\| \nabla \ell\_s(\widehat{\theta}\_s) \\|\_{V\_s^{-1}}^2$, where $\ell_s$ is the Huber loss function and $V_s$ is the local norm matrix. In the presence of heavy-tailed noise, the gradients $\nabla \ell_s(\widehat{\theta}_s)$ can be significantly corrupted by noise, making prior techniques inapplicable. Moreover, the introduction of a normalization factor causes the stability term to grow as $\mathcal{O}(\sqrt{T})$ (as shown in line 316-328 right column), which is undesirable.
> > >
> > > We observe that standard OMD analyses include a negative term typically used to control the generalization gap, and we found that this negative term can also be exploited to partially cancel the $\mathcal{O}(\sqrt{T})$ growth of the stability term. By carefully designing the learning rate of OMD, we are able to effectively eliminate this growth. For more details, please refer to lines 745-751 in Appendix B.3 and lines 898-916 in Appendix B.5.
> > >
> > >
> > > **(ii) Heavy-tailed noise.** The Huber loss may assign a linear penalty even to normal (non-outlier) data, which can reduce its effectiveness. To address this issue, we design a *recursive normalization factor* to ensure that denoised data lies within the quadratic region of the loss function, thereby making better use of clean data.
> > >
> > > Specifically, the normalization factor at round $t$ is set based on the estimation error from the previous round, i.e., $\sigma_t \sim \sqrt{\beta_{t-1}}$ (shown in line 5 in Algorithm 1). Intuitively, when the current estimation error is large, a larger factor $\sigma_t$ helps improve the estimation accuracy. From a theoretical perspective, we leverage a high-probability bound on the estimation error, $\\|\widehat{\theta}\_t - \theta\_\*\\|_{V\_{t-1}} \leq \beta_t$, which ensures that $\left| \left( X\_t^\top \widehat{\theta}\_t - X\_t^\top \theta\_* \right) / \sigma\_t \right| \leq \tau\_t / 2$.
> > >
> > > Additional efforts have also been made to achieve the high-probability bound, which are not expanded upon here. For more details, please refer to lines 634–659 in Appendix B.2, the proof of Lemma 5 in Appendix B.4, and lines 916–961 in Appendix B.5.
> > >
> > > In the revised version, we will include more experiments and improve the presentation to better emphasize the challenges and technical contributions. Thank you again for your valuable feedback!

---

### Decision · Program_Chairs · 2025-05-01

**Decision:**

Accept (poster)

**Comment:**

The paper proposes a novel algorithm, Hvt-UCB, for linear bandit problems with heavy-tailed noise. The algorithm achieves state-of-the-art regret upper bounds while significantly reducing the per-round computation time from O(tlog⁡T) to O(1), and eliminating the need to store historical data. This efficiency is achieved by leveraging the Online Mirror Descent (OMD) framework for online updates of the regression parameters.

The reviewers are generally in agreement about the theoretical challenges involved in reducing computation time for bandit algorithms under heavy-tailed rewards, and they acknowledge the contribution of the work to addressing this gap in the literature.